# Global plant trait relationships extend to the climatic extremes of the tundra biome

H.J.D. Thomas ● et al.#

The majority of variation in six traits critical to the growth, survival and reproduction of plant species is thought to be organised along just two dimensions, corresponding to strategies of plant size and resource acquisition. However, it is unknown whether global plant trait relationships extend to climatic extremes, and if these interspecific relationships are confounded by trait variation within species. We test whether trait relationships extend to the cold extremes of life on Earth using the largest database of tundra plant traits yet compiled. We show that tundra plants demonstrate remarkably similar resource economic traits, but not size traits, compared to global distributions, and exhibit the same two dimensions of trait variation. Three quarters of trait variation occurs among species, mirroring global estimates of interspecific trait variation. Plant trait relationships are thus generalizable to the edge of global trait-space, informing prediction of plant community change in a warming world.

#A full list of authors and their affiliations appears at the end of the paper.

Despite the vast diversity of life on Earth, vascular plants are limited by trade-offs in leaf[1], wood[2], seed[3] and root[4] traits, enabling the characteristics of global plant species to be organised along a few general dimensions[5–8]. Two dimensions, plant size (large and woody vs. small and non-woody) and resource economics (acquisitive vs. conservative), have been shown to describe the majority of variation in six widely-sampled plant traits, which together represent key differences in plant form and function[8]. Such trait relationships predict community assembly[9,10] and ecosystem functions[11,12] across biogeographic gradients[13] and in response to environmental change[12,14]. However, our current understanding of trait relationships has largely been formulated using tropical and temperate data, which comprise over 90% of global trait observations[15] (Fig. 1). Although some site-specific studies exist[16,17], whole-plant trait relationships have not been widely tested at the environmental extremes of plant life such as the cold tundra biome, where plants could exhibit rare or unique trait relationships resulting from adaptation to extreme environmental conditions[18,19].

Our current understanding of global trait relationships is also based on the assumption that the majority of trait variation occurs among species[20]. However, trait variation within communities is ultimately driven by differences among individuals, rather than species[21]. Large within-species trait variation could

thus obscure or alter interspecific trait relationships[22–24], restricting their potential for ecological prediction across scales and among biomes. Within-species variation accounts for approximately 25% of trait variation at the global scale[21], but has been hypothesised to be greater in extreme environments due to environmental filtering of trait expression[25], at local geographical scales where species richness is low[26,27], and for species that span large biogeographical gradients[21] due to wide niche breadth. The tundra biome thus provides an optimal system to test our current understanding of trait variation within plant communities due to a small species pool[28], large species ranges[13], and extreme environmental conditions[29].

In this study, we test whether our existing understanding of plant trait relationships extends to the tundra biome. We establish the largest database of Arctic and alpine tundra plant traits ever compiled by combining 20,991 records from the TRY database[15] with 30,616 records from the Tundra Trait Team (TTT)[30], representing 89% of the tundra species pool. We select six globally well-sampled plant traits: adult plant height, leaf area, seed mass, leaf mass per area (LMA), leaf nitrogen, and leaf dry matter content (LDMC) (Supplementary Table 1). These traits underpin two important dimensions of global trait space[8], and link to ecosystem functions including primary productivity[7], carbon storage[31], and nutrient cycling[12]. We test three

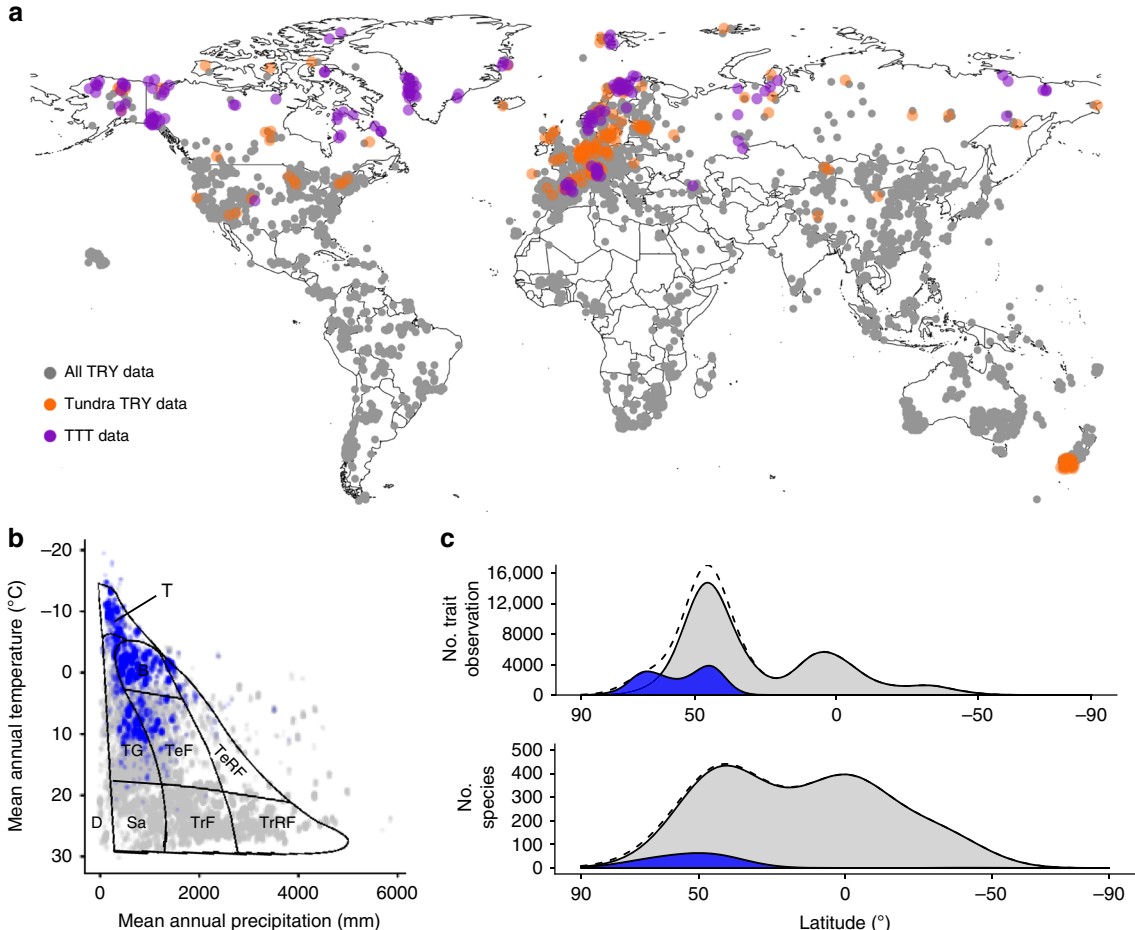

**Fig. 1 Tundra trait data within geographical and climate space. a** Map of trait observation sites for six plant traits, indicating global trait observations in TRY (grey points), tundra species observations in TRY (orange points) and TTT data (purple points). **b** Location of trait collection sites in climate space for all available plant species (grey) and tundra species (blue). Major biomes are mapped onto climate space (T-Tundra; B-Boreal Forest; TG-Temperate Grassland; TeF-Temperate Deciduous Forest; TeRF-Temperate Rain Forest; TrF-Tropical Deciduous Forest; TrRF-Tropical Rain Forest; Sa-Savanna; D–Desert)[146]. **c** Number of trait observations (upper panel) and species (lower panel) for all available plant species (grey) and tundra species (blue), by latitude. Dotted curves indicate global distributions with the inclusion of TTT collected data.

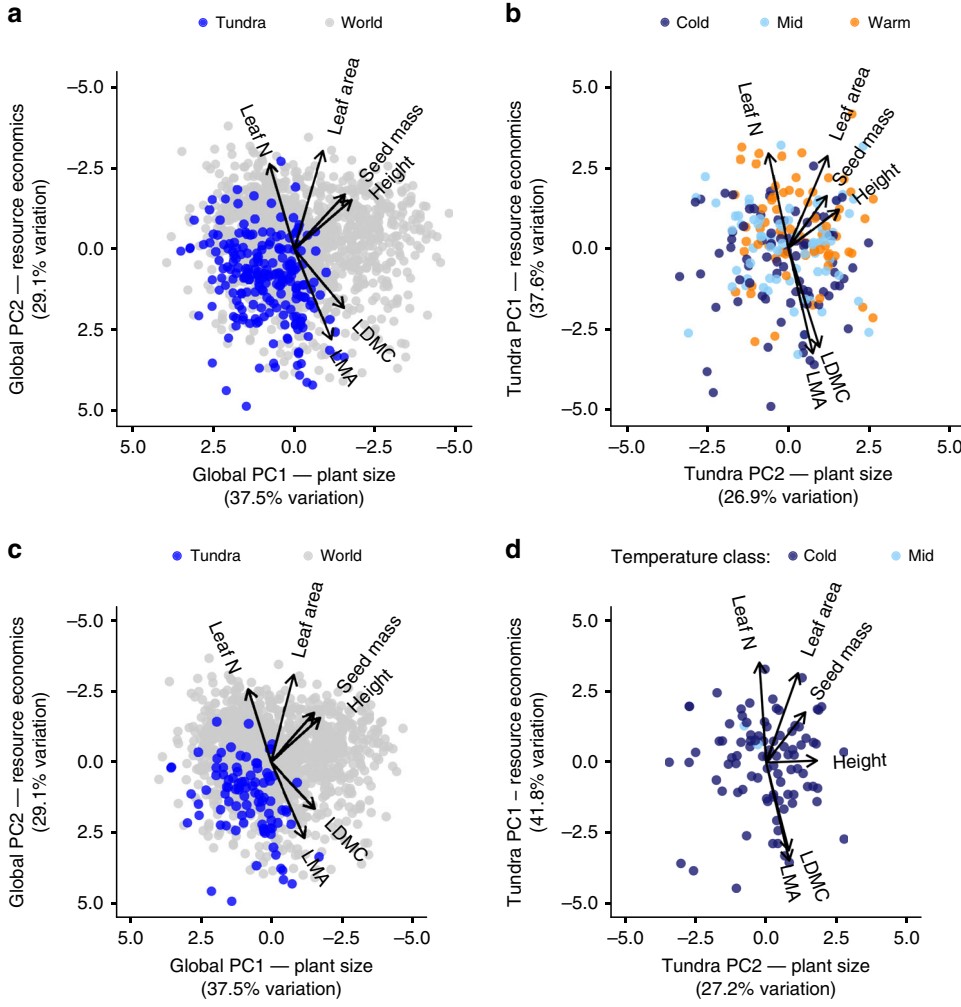

**Fig. 2 Global trait relationships are maintained in the tundra biome despite constrained size, but not resource economic, traits among tundra species.**
**a** Global trait-space defined by six plant traits for 1,358 plant species in the global dataset (grey points) and 219 tundra species (blue points). **b** Distribution of trait space for tundra species only. Note that PCA axes are reversed in tundra data relative to global data. Points are coloured by temperature category, corresponding to the mean annual temperature of trait collection sites for each species (Cold < −1 °C, Mid > −1 °C but <1 °C, Warm >1 °C, Supplementary Fig. 1). Arrows indicate the direction and weighting of trait vectors. We also tested the consistency of the patterns found (**c**) within global trait space and for (**d**) tundra trait space using subset of "extreme" tundra species that included only those species found only north of the Arctic circle or at sites with a MAT < 0 °C.

hypotheses: (1) Trait expression among tundra species will be constrained relative to global trait space due to extreme environmental conditions, yet will exhibit the same two dimensions of plant form and function. (2) The contribution of within-species trait variation to total trait variation in the tundra will be greater than the global average of 25%[21]. (3) The contribution of within-species trait variation to total trait variation will be greater at local rather than at larger geographical scales. We show that tundra plants exhibit constrained size traits, but not resource traits, relative to global species. However, tundra plants exhibit the same two dimensions of trait variation, indicating that plant trait relationships are generalizable to cold extremes of life on Earth. We also show that differences among species comprise the majority of trait variation, but that trait variation within species becomes increasingly important at small geographical scales.

## Results and discussion
**Trait expression is constrained in the tundra**. We found that tundra species occupied a constrained subset of global trait space for size-related traits but not resource economic traits (Fig. 2a, c,

Supplementary Fig. 1). Many tundra species, such as the prostrate, small-leaved, and wind-dispersed evergreen shrub *Cassiope hypnoides*, were located at the very edge of global trait space, consistent with adaptation to extreme environmental conditions in the tundra[32]. Given that tundra plant communities are found above treeline, and therefore by definition exclude tree species, we expected to see reduced plant height among tundra species compared to global species. However, we found that lower plant height corresponded with smaller leaf area and seed mass (Fig. 2a, axis 1, Supplementary Fig. 1), as would be predicted from global trait relationships[5]. In contrast, traits associated with resource economics occupied almost the full global range (Fig. 2a, axis 2), with both highly acquisitive species such as chickweed (*Stellaria media*), and highly conservative species such as crowberry (*Empetrum nigrum*) present at tundra sites. Species with faster resource-related traits and larger size-related traits were associated with warmer environments within the tundra (Fig. 2b, Supplementary Fig. 1), potentially informing the adaptive capacity to climate warming within and among tundra plant communities.

**Global trait relationships extend to the tundra biome**. We found that plant trait relationships among tundra species were consistent with global patterns (Fig. 2b, d), despite a limited range of trait values and lower species richness in the tundra biome. The two dimensions of global trait space (plant size and resource economics) aligned with trait relationships among tundra species (Supplementary Fig. 2), and together explained 64.5% of trait variation in the tundra. However, the relative importance of the PCA axes was reversed relative to global data (Supplementary Fig. 3), suggesting that tundra plant strategies are primarily differentiated by resource economics. Leaf area was more strongly associated with plant size among tundra species and with resource economics among global species (Supplementary Fig. 4). In contrast, leaf dry matter content (LDMC) was more strongly associated with resource economics in the tundra. LDMC correlated closely with stem density, which was associated with plant size and structure among global plant species[8], especially among tree species[2]. Nevertheless, trait co-variation was maintained in the tundra despite the absence of trees, which comprise half of global trait space[8] and have been a focus of many previous studies of plant trait relationships[1,2,5,6].

**The majority of trait variation occurs among species**. We found that differences among species explained the majority of trait variation in the tundra biome, accounting for an average of 76.8% of variation across the six traits examined (Fig. 3a, Fig. 4) and reinforcing one of the key assumptions of trait-based ecology[20]. Functional group categorisation alone explained an average of 25.6% of trait variation across all six traits, but varied substantially by trait;[33] differences among species still accounted for the majority of trait variation even if functional group classifications were removed. The contribution of within-species variation to total trait variation (23.2%) was surprisingly close to the global mean (25%[21]), despite harsh environmental conditions and large species ranges in the tundra. However, within-species variation differed substantially by trait, accounting for as much as 55% of trait variation for leaf nitrogen, in line with previous studies[15,21]. Size-related traits demonstrated greater overall variation than resource economic traits, even though variation relative to global trait space was constrained along the size-related axis (Fig. 2a). Overall, our findings support the hypothesis that species-level variation comprises the majority of the global spectrum of plant form and function[8,20], underlining the importance of species richness and turnover in determining plant community characteristics, trait diversity, and linkages to ecosystem function.

**Trait variation across geographic scales**. We found that the contribution of within-species trait variation was largely consistent across geographic scales (Fig. 4a–d), but comprised a greater proportion of total variation at local scales (<10 km[2]), approximately the size of current high-resolution cells in gridded climate datasets. Sites with low sampled species richness also exhibited high within-species variation (<10 species; Fig. 4e-h), suggesting that spatial patterns were at least in part driven by a small species pool at local scales. Although both theoretical models[27] and empirical studies[21] have suggested that the contribution of within-species trait variation should increase at local scales, we demonstrate this relationship from the plot to the biome scale.

Our findings indicate that the relationships between plant traits found at the global scale are generalizable even at the climatic edge of global trait-space. Our results suggest that plants are subjected to globally consistent trade-offs in trait expression[7,8,16] despite dramatically different environmental constraints[17], growth forms[29] and evolutionary history[6] across biomes. Our findings reinforce claims that relationships between these widely measured plant traits are indicative of fundamental trade-offs in plant life strategy[8,9], including resource acquisition (LMA, LA, LN, LDMC)[7], competition (PH, SM, LA, LDMC)[34], and reproduction (PH, SM)[5]. However, plant size and resource economics have yet to be integrated with other key facets of plant life strategy such as phenological[35,36], chemical and belowground traits[4,18]. These less frequently measured traits need to be incorporated into analyses to more comprehensively capture how extreme biomes such as the tundra occupy global trait space.

Tundra plant species showed remarkable variation in resource economic traits within the tundra biome relative to global trait space[8]. Given the low vascular plant diversity associated with many tundra environments, this variation in plant leaf resource economics is notably high and suggests that tundra species have developed a wide range of ecological strategies to cope with extreme conditions and limiting resources. In contrast, tundra plant species occupied half the global range of size-related traits, potentially indicating that two of the major axes of global trait variation may be differentially selected by environmental

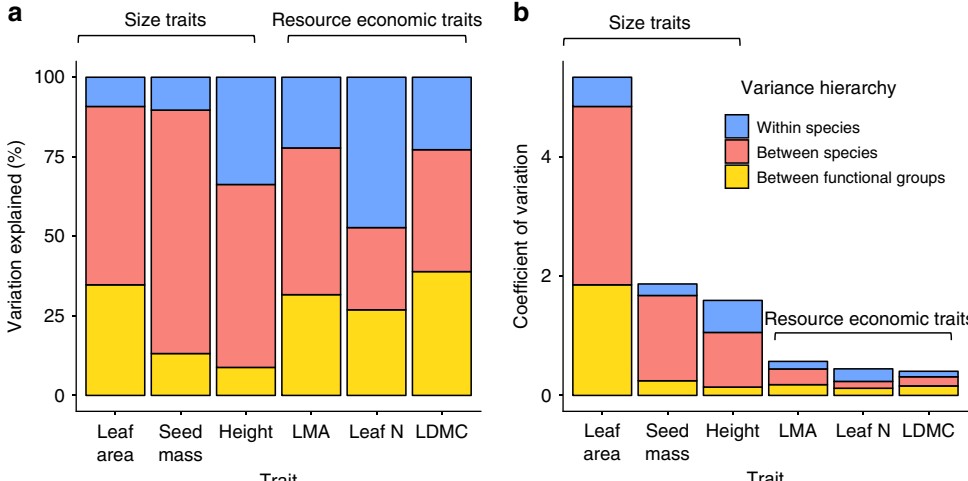

**Fig. 3 Sources of trait variation for six plant traits in the tundra biome. a** Relative proportion of trait variation explained by functional group (deciduous shrubs, evergreen shrubs, graminoids, forbs; yellow), species (red) and within species (blue). **b** Total trait variation, represented by the coefficient of variation (ratio of the standard deviation to the mean), and component sources of trait variation.

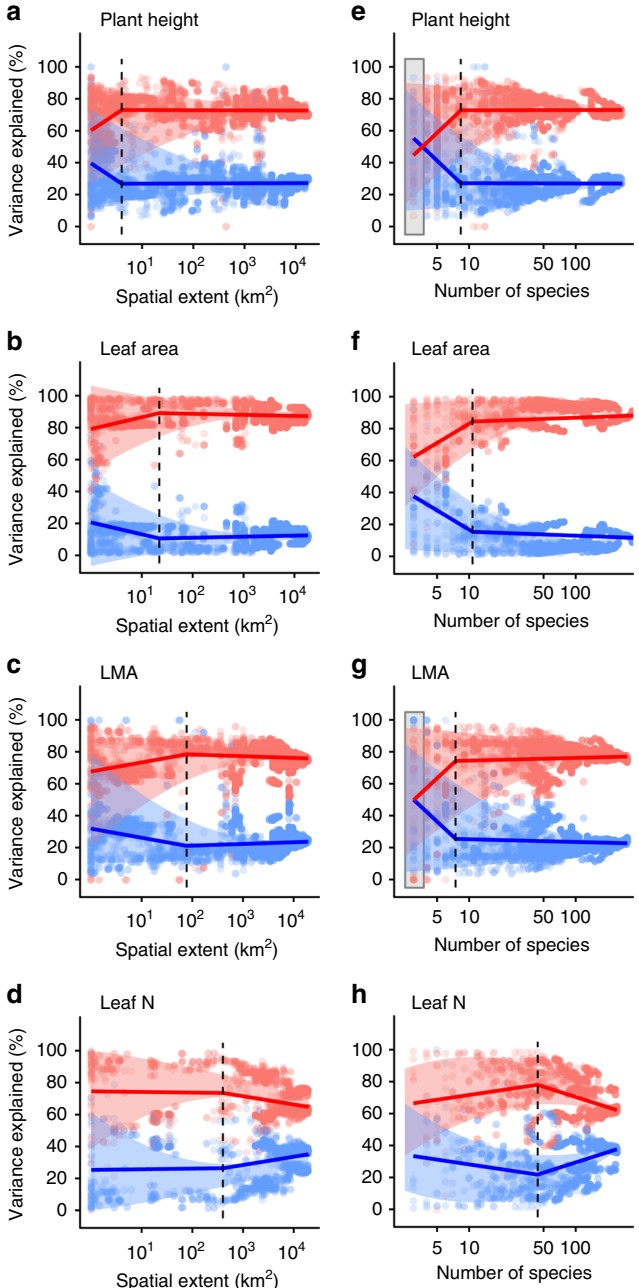

**Fig. 4 Sources of trait variation across geographic scales.** Among-species trait variation (red) accounts for the majority of total trait variation across the tundra, but the importance of within-species trait variation (blue) increases at local scales and at low species richness in most traits. Sources of trait variation across geographical scale (**a–d**) and species richness (**e–h**) for plant height, leaf area, LMA and leaf nitrogen (see also Supplementary Figs. 5–7). Coloured lines indicate linear break point model fits with one break point (dashed line). Grey boxes indicate where differences between among-species and within-species variation are not significant ($P > 0.05$).

conditions, and could thus respond differently to environmental change. If the spatial patterns observed in our study are indicative of temporal responses[13,37], climate change will likely shift trait distributions towards increased plant height, leaf area and seed mass, as has already been observed at some sites[38,39], and for plant height at the tundra biome scale[13].

Contrary to our expectations, the contribution of within-species trait variation in the tundra was lower than global

estimates[21]. Our findings support previous studies indicating that the relative importance of within-species variation decreases with increasing environmental stress[25], leading to wide functional divergence between species, as found for resource economic traits[33,40]. Lower within-species variation may further indicate that plasticity is lower among tundra species, which are typically slow growing and nutrient limited[41]. However, rapid and sustained plastic responses to environmental change have been documented at some tundra sites[38,39]. Indeed, large differences in both trait expression, as demonstrated here, and plasticity have been found to promote coexistence in other resource-limited comminities[36]. If the majority of trait variation occurs among species, and if phenotypic plasticity is comparatively low, shifts in community-level traits following environmental change may occur more slowly than would be predicted from biogeographic gradients[13,37]. More substantial trait change would thus require the immigration of new species from warmer sites.

We found that within-species trait variation comprised a large component of trait variation at local scales—the scale at which many critical ecological processes occur[11]. Despite the importance of trait differences among species in the tundra, we nevertheless found that that within-species variation accounted for approximately one quarter of total trait variation, and thus should not be ignored in trait-based analyses[42]. High within-species trait variation at local scales has previously been predicted from ecological theory[21,27], and may result from low local-scale species richness[27] or reveal the influence of local-scale environmental variability (i.e., topography, snow, drainage, etc.)[24,43–45]. Accounting for multiple sources of trait variation has been shown to constrain trait-based vegetation models[7,12] and subsequently to improve prediction of the response of key ecosystem processes to environmental change[11,13,46]. However, such trait-based modelling approaches are rare, and require precisely geo-referenced trait databases that link trait records to environmental variables. We therefore support calls to collect additional trait data in changing and novel climate conditions[13,30], to improve the techniques and technologies used to remotely sense plant trait information[28], and to incorporate trait variation into Earth system modelling[47].

Overall, our findings demonstrate that relationships and trade-offs among six fundamental plant traits are generalisable across lineages and among biomes to the cold extremes of the planet. It remains to be tested whether such relationships are consistent in other environmental extremes, such as desert ecosystems. Understanding the differences in trait expression across global trait-space offers fundamental insights into the rules that underpin evolution, community assembly, and ecosystem response to environmental change. Quantifying trait expression and variation thus offers a clearer picture of how plant communities and ecosystem functions will respond as climate change alters environmental conditions around the world.

## Methods
**Tundra biome definition**. In line with previous biome-scale assessments of tundra vegetation community change[48–50], we defined the tundra biome as the vegetated regions above treeline at high latitude and high altitude. Tundra species were identified as those present in sampling plots from two biome-scale experiments, the International Tundra Experiment (ITEX)[51] and associated sites[49], and the sUM-MITDiv network[52], or those present at trait collection sites with a mean annual temperature below 0ºC. Tundra plant communities include many widely-distributed and locally common species that are found across large geographical gradients and a variety of environments[51]. We included trait records of tundra species collected outside of tundra environments in this study because i) we were specifically interested in the maximum potential within-species variation among tundra species, ii) defining tundra environments on a purely climatic basis (excluding biotic community in the definition) is very difficult and would require arbitrary decisions regarding biome boundaries, and iii) many trait records in the TRY database do not contain georeferenced collection coordinates and thus would be impossible to classify based on environment (Fig. 1b).

**Trait selection**. We selected six plant traits: plant height (PH, maximum measured height), seed mass (SM, dry mass), leaf area per leaf (LA, fresh leaf area), leaf mass per area (LMA, ratio of leaf dry mass to fresh leaf area), leaf dry matter content (LDMC, ratio of leaf dry mass to fresh leaf mass), and leaf nitrogen (LN, nitrogen per unit leaf dry mass). These six traits are considered to represent fundamental dimensions of ecological strategy[8,53] and are commonly measured in the tundra[30], thus maximising trait coverage.

**Trait data collection**. We extracted trait data from the TRY[15] 3.0 database (available at www.try-db.org) for tundra species[1,2,17,54–142]. We extracted traits of all tundra species from the TRY database regardless of location to maximise the capture of trait variation per species. We supplemented TRY data with additional trait data from the "Tundra Trait Team" (TTT) database[30]. All species names from ITEX, TRY and TTT were matched to accepted names in The Plant List using the R package Taxonstand (v. 1.8) before merging the datasets[143]. We assigned species to four traditional functional groups—evergreen shrubs, deciduous shrubs, graminoids, and forbs—based on previous classification of ITEX species[49]. We excluded stem-specific density (SSD) from all analyses, except for comparisons with Diaz et al.[8] (Supplementary Figs. 9–11) since SSD had low collection coverage in the tundra and was available for too few species ($n = 52$). We also extracted trait data and collection site coordinates from TRY 3.0 for all global species to provide global context in geographical-climate-space and trait-space analyses following previously published approaches in Kattge et al.[15] and Diaz et al.[8].

**Data cleaning-TRY**. TRY trait data were subjected to a multi-step cleaning process. Firstly, all values that did not represent individual measurements or species means were excluded. Secondly, we identified overlapping datasets within TRY and removed duplicate observations whenever possible. The following datasets were identified as having partially overlapping observations: GLOPNET–Global Plant Trait Network Database, The LEDA Traitbase, Abisko and Sheffield Database, Tundra Plant Traits Database, and KEW Seed Information Database (SID).

Thirdly, we removed duplicates within each TRY dataset (e.g., if a value is listed once as "mean" and once as "best estimate") by first calculating the ratio of duplicated values within each dataset, and then removing duplicates from datasets with more than 30% duplicated values. This cut-off was determined by manual evaluation of datasets at a range of thresholds. Datasets with fewer than 30% duplicated values were not cleaned in this way as any internally duplicate values were assumed to be true duplicates (i.e., two different individuals were measured and happened to have the same measurement value).

Finally, we removed all species mean observations from the "Niwot Alpine Plant Traits" database and replaced them with the original individual observations as provided by the trait collector (Marko J. Spasojevic) in order to ensure all trait measurements were for individuals.

**Data cleaning–TRY and TTT combined**. Both datasets were checked for improbable values, with the goal of excluding likely errors or measurements with incorrect units, but without excluding true extreme values. It was particularly important to avoid artificial reduction in the range of trait values in this study, since we were explicitly interested in trait variation. We followed a series of data-cleaning steps, in each case estimating an error risk for a given observation (x) by calculating the difference between x and the mean (excluding x) of the group in question and then dividing by the standard deviation of the group. We employed a hierarchical data cleaning method, because the standard deviation of a trait value is related to the mean and sample size. First, we checked individual records against the entire distribution of observations of that trait and removed any records with an error risk greater than 8.0 (i.e., a value more than eight standard deviations away from the trait mean). For species that occurred in four or more unique datasets within TRY or TTT (i.e., different data contributors), we estimated a species mean per dataset and removed observations for which the species mean error risk was greater than 3.0 (i.e., the species mean of that dataset was more than three SD's away from the species mean across all datasets). For species that occurred in fewer than four unique datasets, we estimated a genus mean per dataset and removed observations in datasets for which the error risk based on the genus mean was greater than 3.5. Finally, we compared individual records directly to the distribution of values for that species. For species with fewer than four records, we did not remove any values. For species with more than four records, we excluded values above an error risk y, where y was dependent on the number of records of that species and ranged from an error risk of 2.25 for species with fewer than 10 records to an error risk of 4.0 for species with more than 30 records. This procedure was performed on the complete tundra trait database – including species and traits not presented here. In total 3515 observations (2.8%) were removed. In all cases, we visually checked the excluded values against the distribution of all observations for each species to ensure that our trait cleaning protocol was reasonable.

All trait observations with latitude/longitude information were mapped using the R package 'mapproj'[144] and checked for illogical values (e.g., falling in the ocean). These values were corrected from the original publications or by contacting the data contributor whenever possible. Where locations could not be verified, geo-referenced coordinates were removed and the trait data not included in geographic analyses.

**Final trait database**. After cleaning out duplicates and suspected mistakes, we retained 51,657 unique trait observations (of which 20,991 were already in TRY and 30,616 were newly contributed by the Tundra Trait Team) across the six traits of interest. Of the 447 identified species in the ITEX dataset, 397 (89%) had trait data available from TRY or TTT for at least one trait (range 60–100% per site). Those species without trait data generally represent rare or uncommon species unique to each site. On average, trait data were available for 97% of total plant cover across all sites (range 39–100% per site; Supplementary Table 1).

Data compiled through the Tundra Trait Team are available in Bjorkman et al.[30]. The total TTT database submitted to TRY includes traits not considered in this study, as well as tundra species that do not occur in our vegetation survey plots, for a total of 54,210 trait observations on 530 species. For more information on trait data and trait cleaning methods see Bjorkman et al.[13] and Bjorkman et al.[30].

**Climate data**. To plot the distribution of tundra trait data within climate space, we used the coordinates of all unique collection sites for both tundra (TRY and TTT) and global (TRY) datasets.

CHELSA climate variables (mean annual temperature–BIO10_1 and mean annual precipitation–BIO10_12, http://chelsa-climate.org/) were extracted for all trait observations with latitude/longitude values recorded (39,573 records in total, 12,434 of which were from TRY and 27,139 from TTT). Because most observations did not include information about elevation, temperature estimates for individual trait observations were not corrected for elevation.

We calculated the 'temperature class' of tundra species based on the mean summer temperature of trait collection sites for each species. Mean summer temperature was considered to be the most ecologically meaningful climatic variable since it captures conditions during the growing season for each plant species. We extracted summer temperatures from the CHELSA dataset based on BioClim variable BIO10_1. We assigned species to three temperature classes: Cold tundra = mean summer temperature less than $-1\,°C$, Mid tundra = mean summer temperature greater than $-1\,°C$ but less than $1\,°C$, Warm tundra = summer temperature of coldest site greater than $1\,°C$.

**Analysis of trait relationships**. All analyses were conducted in R (v. 3.3.3). Code is available at github.com/hjdthomas/Tundra_trait_variation.

We performed principal component analysis (PCA) on plant traits for all global species, and for tundra species only using the R package 'prcomp'. As far as possible, we replicated the methods outlined in Diaz et al. (2016)[8], though this was not always possible due to the use of gap-filled traits and additional data not included in TRY in Diaz et al. (2016)[8]. We log transformed trait values to account for log-normal distributions, which is considered appropriate for data with different measurement scales[8]. To test whether the inclusion of SSD altered results, and for comparison with Diaz et al. (2016)[8], we performed supplementary analysis using a conversion from SSD to LDMC based on the correlation between these two traits[8,145], since LDMC and SSD individually have fewer trait observations than other traits at the global scale. Although only an approximate conversion, this greatly increases the number of species available for the analyses and does not affect the distribution of trait-space or direction of trait loadings (Supplementary Figs. 8–10). We did not use converted values in the main analysis to avoid introducing additional sources of variation.

To visualise trait-space, we plotted the first two PCA axes and direction and weighting of trait loadings. We performed PCA on the full global dataset (including tundra data) and highlighted tundra species within the overall distribution to identify the location of tundra species within global trait-space. We repeated PCA using tundra species only to compare global trait relationships with tundra trait relationships. We compared all pairwise trait correlations for both global and tundra species (Supplementary Fig. 2) to investigate consistency in trait-trait relationships, and investigated the strength and direction of trait loadings for global and tundra analyses to compare the location and relative importance of PCA axes (Supplementary Fig. 3). Finally, we calculated the contribution of traits loadings to each PCA axis using the 'fviz_contrib' function in the R package 'factoextra' (Supplementary Fig. 4).

To investigate whether the location of species within tundra trait-space or along the two major axes of variation was influenced by climate, we categorised species according to temperature class (point colour). We plotted the distribution of species along PCA axes for each temperature class to test whether trait variation within multivariate trait-space was influenced by the species' thermal range (Supplementary Fig. 1).

**Variance partitioning**. To investigate the sources of trait variation in the tundra, we conducted variance partitioning by fitting a generalised linear mixed-effects model to the variance across nested classification hierarchy (functional group/ species) using the R package 'nlme'. We then conducted a variance component analysis on this model using the 'varcomp' function in the R package 'ape'. Partitioning was performed on a trait-by-trait basis, so does not account for co-variation between traits. We used unexplained variance to represent the within-species variation (including within-individual variation), though some

unexplained variation could also arise from measurement error. Variance partitioning was conducted for each site with greater than three observations per trait per species, and at least three sampled species, and summarised using the mean of all sites. To complement variance partitioning, we also calculated the coefficient of variation (CV; the ratio of the standard deviation to the mean) to compare variation among traits. CV was calculated for each trait for all species.

**Variance partitioning across geographic scale**. To assess how variance explained by differences within-species and between-species varied with geographic scale, we iteratively grouped sites based on geographic proximity. We calculated the geographic distance between all trait sampling sites using the R package 'geosphere'. We excluded sampling sites with fewer than three species per site, or fewer than two trait observations per species. To test the sensitivity of findings to trait or species availability, we also conducted a sensitivity analysis excluding sites with fewer than five species per site or trait observations per species (Supplementary Figs. 12, 13). For a given trait sampling site, we conducted variance partitioning analysis at the site scale (scale = 0). We then added data from the nearest site (shortest geographical distance) and conducted variance partitioning analysis on this expanded dataset (geographic scale = distance from starting site to most distant site). We added sites iteratively until all sites were included i.e., the biome-scale was reached. We repeated this analysis across all trait sampling sites. To examine whether sources of trait variance were affected by differences in species richness at different geographical scales, we also calculated the species richness (number of unique species for which we have trait measurements, i.e., size of the measured species pool) of the dataset at each sampling step.

To summarise the relationship between variance explained, geographic scale and species richness, we performed a breakpoint analysis using the R package 'segmented' with one break point. To calculate errors, we grouped trait observation sites into 5 km or 1 species bins and calculated the 95% intervals of the spread of values. These were plotted as error bounds using a second-order polynomial smooth. We visualised all analyses with geographic scale and species richness presented on a $log_{10}$ scale to highlight change at local scales. For non-logged figures see Supplementary Figs. 6, 7. To investigate if differences in the contribution of within-species and among-species variation to total trait variation were significant, we additionally grouped data into 10 equal bins, and tested the significance of source of variance using linear models, with a significance threshold of 0.05. We highlighted insignificant bins on figures as grey shaded areas, which indicate scales at which the amount of within-species variation is not significantly different from among-species variation. We examined (1) the mean contribution of within-species and among-species variation to overall trait variation, and (2) proportion of sampling combinations for which within-species variation accounted for at least one third and one half of overall trait variation above and below each break point (Supplementary Fig. 5). Finally, we tested the sensitivity of all analysis to species selection by repeating analyses for only trait sampling sites north of the Arctic Circle (66.5 °N) or from collection locations with a mean annual temperature below 0 °C (Supplementary Figs. 14–20).

**Reporting summary**. Further information on research design is available in the Nature Research Reporting Summary linked to this article.

## Data availability
All trait data compiled through the Tundra Trait Team are available at http://github.com/TundraTraitTeam/TraitHub, see Bjorkman et al.[30]. Additional global trait data are available through the TRY database (www.try-db.org)[15].

## Code availability
Code is available at http://github.com/hjdthomas/Tundra_trait_variation.

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

## Acknowledgements

Additional data and contributions were provided by L. Andreu-Hayles, O. Atkin, A. Blach Overgaard, J. Dickie, S. Dullinger, B. Enquist, J. Fang, K. Fleischer, H. Ford, G. Freschet, E. Garnier, R. Halfdan Jørgensen, K. Harper, S. Harrison, M. Harze, J. Hille Ris Lambers, R. Jackson, R. Klady, S. Kuleza, A. Lavalle, F. Louault, B. Medlyn, R. Milla, J. Ordonez, C. Pladevall, H. Poorter, C. Price, P. Semenchuk, F. Schweingruber, B. Shipley, A. Siefert, L. Street, J. Tremblay, E. Weiher, C. Wirth, I. Wright and the Royal Botanic Gardens Kew Seed Information Database (SID). We thank innumerable field technicians, logistics teams, graduate and undergraduate assistants for help with data

collection, and parks, wildlife refuges, field stations and the local and indigenous people for the opportunity to conduct research on their land. Thanks to all those that have reviewed this manuscript for their insightful and constructive comments. The project was funded by the UK Natural Environment Research Council (ShrubTundra Project NE/M016323/1 [I.M.-S., H.T., A.B.] and NERC doctoral training partnership grant NE/L002558/1 [H.T.]), and the Synthesis Centre of the German Centre for Integrative Biodiversity Research (iDiv) Halle-Jena-Leipzig (DFG FZT 118; sTundra working group). The study has been supported by the TRY initiative on plant traits (http://www.try-db. org). The TRY initiative and database is hosted at the Max Planck Institute for Biogeochemistry, Jena, Germany. TRY is currently supported by DIVERSITAS/Future Earth and the German Centre for Integrative Biodiversity Research (iDiv) Halle-Jena-Leipzig. Authors were supported by: Academy of Finland [A.E.], ArcticNet [E.F., E.L., L.H., O.G.], Arctic Research Centre, Aarhus University [J.N.N.], BBSRC David Phillips Fellowship [F.d.V.], Carl Tryggers Stiftelse för Vetenskaplig Forskning and Qatar Petroleum (QUEX-ESC-QP-RD-18/19 [J.M.A.], Carlsberg Foundation [S.N., S.S.N.], Centre d'études nordiques [O.G.], Danish Council for Independent Research-Natural Sciences [S.N., S.S.N.], Danish National Research Foundation [B.E.], Deutsche Forschungsgemeinschaft DFG [N.R., M.D.], Energy Exascale Earth System Model (E3SM) project, funded by the U.S. Department of Energy, Office of Science, Office of Biological and Environmental Research [B.B.-L.], EU-F7P INTERACT [A.B.], EU-INTERACT [MH], European Research Council Synergy Grant ERC-2013-SyG-610028 IMBALANCE-P [J.P.], Fonds de recherche du Quebec: Nature et technologies [O.G.], Marie Skłodowska Curie Actions [D.B.], MOBILITY PLUS [A.B.], Montagna di Torricchio Nature Reserve [G.C.], National Aeronautics and Space Administration [S.S.], NASA's Arctic Boreal Vulnerability Experiment (ABoVE) [S.J.G., L.T.B.], Natural Environment Research Council (UK) [NE/M019160/1; B.B.], Natural Sciences and Engineering Research Council of Canada [B.S., E.F., E.L., J.J., M.V., P.M., O.G., P.G., T.Z.], National Science Foundation (USA) [K.G., R.H.; Arctic Natural Sciences program S.J.G, L.T.B], Netherlands Organization for Scientific Research [R.S.], The Niwot Ridge LTER (NSF DEB-1637686) [M.J.S, K.N.S.], Northern Scientific Training Program [O.G.], NWO Earth and Life Sciences (NWO-ALW), project ALWPP.2016.008 [MH] Memorial University [L.H.], Organismo Autónomo Parques Nacionales [J.M.N.], Polar Continental Shelf Program [E.F., E.L., O.G.], Research Council of Norway [J.S.], Russian Science Foundation, #16-14-10208 [V.O.], Swedish Research Council [D.B., E.K., R.B.], Swiss National Science Foundation [A.K., E.F.], University of Zurich Research Priority Program on Global Change and Biodiversity [G.S.S.], U.S. Department of Energy, Office of Biological and Environmental Research-Energy Exascale Earth System Model [B.B.L.], U.S. Department of Energy, Office of Biological and Environmental Research-Next-Generation Ecosystem Experiments in the Arctic (NGEE Arctic) [C.I.], University of Zurich Research Priority Program on Global Change and Biodiversity [M.I.G.], and the Villum Foundation [S.S.N.]. Any use of trade, firm, or product names is for descriptive purposes only and does not imply endorsement by the U.S. Government.

## Author contributions
H.T., I.M.-S. and A.B. conceived the study. H.T. performed statistical analysis with additional input from I.M.-S. and A.B. H.T. wrote the manuscript with input from I.M.-S. and A.B., and contributions from all authors. A.B. compiled the TTT database with assistance from I.M.-S., H.T. and S.A.B. H.T., A.B., I.M.-S., S.E., J.K., S.D., M.V., D.B., J.C., B.F., G.H., R.H., S.N., J.P., C.R., G.S.S., M.W., S.W., W.C., P.B., D.G., S.G., K.G., N.R., N.S., M.S., J.A., H.A., A.A.R., S.A.B., Mt.B., L.B., R.B., A.B., A.B., M.C., K.C., L.C., E.C., B.E., A.E., E.F., O.G., P.G., M.H., L.H., J.H., J.J., K.H., M.I.G., C.I., F.J.,. E.K., A.K., L. J.L., T.C.L., E.L., C.L., A.M., A.M., J.N.N., S.N., J.N., S.O., J.O., V.O., A.P., S.R., R.S., J.S., K.S., K.T., M.T., A.T., U.T., M.T., S.V., T.V., S.W., P.W., T.Z., M.B., B.B., P.v.B., B.B.L., G.C., B.C., F.S.C. III, J.C., M.D., W.G., S.J., M.K., P.M., U.N., Y.O., W.O., J.P., P.P., P.R., B.S., S.S. and F.d.V. contributed data. I.M.-S., S.E. and A.B. led the sTundra working group. I.M.-S. supervised H.T. and acquired funding for the project.

## Competing interests
The authors declare no competing interests.

## Additional information

H.J.D. Thomas [1⬚], A.D. Bjorkman [1,2,3], I.H. Myers-Smith [1], S.C. Elmendorf [4], J. Kattge [5,6], S. Diaz [7,8], M. Vellend [9], D. Blok [10], J.H.C. Cornelissen [11], B.C. Forbes [12], G.H.R. Henry [13], R.D. Hollister [14], S. Normand [15], J.S. Prevéy [16,17], C. Rixen [17], G. Schaepman-Strub [18], M. Wilmking [19], S. Wipf [17,20], W.K. Cornwell [21], P.S.A. Beck [22], D. Georges [1,23], S.J. Goetz [24], K.C. Guay [25], N. Rüger [6,26], N.A. Soudzilovskaia [27], M.J. Spasojevic [28], J.M. Alatalo [29,30], H.D. Alexander [31], A. Anadon-Rosell [19,32,33], S. Angers-Blondin [1], M. te Beest [34,35], L.T. Berner [24], R.G. Björk [36,37], A. Buchwal [38,39], A. Buras [40], M. Carbognani [41], K.S. Christie [42], L.S. Collier [43], E.J. Cooper [44], B. Elberling [45], A. Eskelinen [6,46,47], E.R. Frei [13,48], O. Grau [49,50,51], P. Grogan [52], M. Hallinger [53], M.M.P.D. Heijmans [54], L. Hermanutz [43], J.M.G. Hudson [55], J.F. Johnstone [56], K. Hülber [57], M. Iturrate-Garcia [18], C.M. Iversen [58], F. Jaroszynska [17,59,60], E. Kaarlejarvi [33,61,62], A. Kulonen [17], L.J. Lamarque [63], T.C. Lantz [64], E. Lévesque [63], C.J. Little [18,65], A. Michelsen [45,66], A. Milbau [67], J. Nabe-Nielsen [68], S.S. Nielsen [15], J.M. Ninot [32,33], S.F. Oberbauer [69], J. Olofsson [35], V.G. Onipchenko [70], A. Petraglia [41], S.B. Rumpf [57,71], R. Shetti [19], J.D.M. Speed [72],

K.N. Suding[4], K.D. Tape[73], M. Tomaselli [41], A.J. Trant[74], U.A. Treier[15], M. Tremblay[63], S.E. Venn [75], T. Vowles[36], S. Weijers [76], P.A. Wookey [77], T.J. Zamin[52], M. Bahn[78], B. Blonder[79,80,81], P.M. van Bodegom [27], B. Bond-Lamberty [82], G. Campetella [83], B.E.L. Cerabolini [84], F.S. Chapin III[85], J.M. Craine [86], M. Dainese [87,88], W.A. Green[89], S. Jansen [90], M. Kleyer [91], P. Manning [92], Ü. Niinemets[93], Y. Onoda [94], W.A. Ozinga [95], J. Peñuelas [49,50], P. Poschlod[96], P.B. Reich [97,98], B. Sandel[99], B.S. Schamp [100], S.N. Sheremetiev[101] & F.T. de Vries [102]

[1]School of Geosciences, University of Edinburgh, Edinburgh, EH9 3FF Scotland, UK. [2]Department of Biological and Environmental Sciences, University of Gothenburg, Medicinaregatan 18, 40530 Gothenburg, Sweden. [3]Gothenburg Global Biodiversity Centre, Carl Skottsbergs gata 22B, 41319 Gothenburg, Sweden. [4]Institute of Arctic and Alpine Research, University of Colorado, Boulder, CO 80309-0450, USA. [5]Max Planck Institute for Biogeochemistry, 07701 Jena, Germany. [6]German Centre for Integrative Biodiversity Research (iDiv), Halle-Jena-Leipzig, Deutscher Platz 5e, 04103 Leipzig, Germany. [7]Instituto Multidisciplinario de Biología Vegetal (IMBIV), CONICET, Av.Velez Sarsfield 299, Cordoba, Argentina. [8]FCEFyN, Universidad Nacional de Córdoba, Av. Vélez Sarsfield 299, X5000JJC, Córdoba, Argentina. [9]Département de Biologie, Université de Sherbrooke, 2500, boul. de l'Université Sherbrooke, Québec, J1K 2R1, Canada. [10]Dutch Research Council, (NWO), Postbus 93460, 2509 AL Den Haag, The Netherlands. [11]Systems Ecology, Department of Ecological Science, Vrije Universiteit, De Boelelaan 1085, 1081 HV Amsterdam, The Netherlands. [12]Arctic Centre, University of Lapland, 96101 Rovaniemi, Finland. [13]Department of Geography, University of British Columbia, 1984 West Mall, Vancouver, V6T 1Z2, Canada. [14]Biology Department, Grand Valley State University, 1 Campus Drive, 3300a Kindschi Hall of Science, Allendale, Michigan, USA. [15]Department of Biology, Aarhus University, Ny Munkegade 114-116, DK-8000 Aarhus C, Denmark. [16]U.S. Geological Survey, Fort Collins Science Center, Fort Collins, CO 80526, USA. [17]WSL Institute for Snow and Avalanche Research SLF, Flüelastrasse 11, 7260 Davos Dorf, Switzerland. [18]Department of Evolutionary Biology and Environmental Studies, University of Zurich, Winterthurerstrasse 190, 8057 Zurich, Switzerland. [19]Institute of Botany and Landscape Ecology, Greifswald University, Soldmannstraße 15, 17487 Greifswald, Germany. [20]Swiss National Park, Runatsch 124, Chastè Planta-Wildenberg, 7530 Zernez, Switzerland. [21]Ecology and Evolution Research Centre, School of Biological, Earth and Environmental Sciences, University of New South Wales, Sydney, NSW 2052, Australia. [22]European Commission, Joint Research Centre, Via Enrico Fermi, 2749, Ispra, 21027, Italy. [23]International Agency for Research in Cancer, 150 Cours Albert Thomas, 69372 Lyon, France. [24]School of Informatics, Computing and Cyber Systems, Northern Arizona University, Flagstaff1295S Knoles DrAZ 86011, USA. [25]Bigelow Laboratory for Ocean Sciences, 60 Bigelow Dr, East Boothbay, Maine, 04544, USA. [26]Smithsonian Tropical Research Institute, Luis Clement Avenue, Bldg. 401 Tupper, Balboa Ancón, Panama. [27]Environmental Biology Department, Institute of Environmental Sciences, Leiden University, 2300 RA Leiden, The Netherlands. [28]Department of Evolution, Ecology, and Organismal Biology, University of California Riverside, Life Sciences Building, Eucalyptus Dr #2710, Riverside, CA 92521, USA. [29]Department of Biological and Environmental Sciences, College of Arts and Sciences, Qatar University, P.O. Box 2713, Doha, Qatar. [30]Environmental Science Center, Qatar University, Doha, Qatar. [31]Department of Forestry, Forest and Wildlife Research Center, Mississippi State University, Mississippi, MS 39762, USA. [32]Department of Evolutionary Biology, Ecology and Environmental Sciences, University of Barcelona, Diagonal, 643, 08028 Barcelona, Spain. [33]Biodiversity Research Institute, University of Barcelona, Av. Diagonal, 645, 08028 Barcelona, Spain. [34]Environmental Sciences, Copernicus Institute of Sustainable Development, Utrecht University, Heidelberglaan 8, 3584 CS Utrecht, The Netherlands. [35]Department of Ecology and Environmental Science Umeå University, SE-901 87 Umeå, Sweden. [36]Department of Earth Sciences, University of Gothenburg, 405 30 Gothenburg, Sweden. [37]Gothenburg Global Biodiversity Centre, SE-405 30 Gothenburg, Sweden. [38]Adam Mickiewicz University, Institute of Geoecology and Geoinformation, B. Krygowskiego 10, 61-680 Poznan, Poland. [39]University of Alaska Anchorage, 3211 Providence Dr, Anchorage, AK 99508, USA. [40]Land Surface-Atmosphere Interactions, Technische Universität München, Hans-Carl-von-Carlowitz Platz 2, 85354 Freising, Germany. [41]Deptartment of Chemistry, Life Sciences and Environmental Sustainability, University of Parma, Parco Area delle Scienze, 11/a, 43124 Parma, Italy. [42]Alaska Department of Fish and Game, 333 Raspberry Rd, Anchorage, AK 99518, USA. [43]Department of Biology, Memorial University, St. John's, Newfoundland and Labrador, A1C 5S7, Canada. [44]Deptartment of Arctic and Marine Biology, Faculty of Bioscences Fisheries and Economics, UiT-The Arctic University of Norway, Tromsø, Norway. [45]Center for Permafrost (CENPERM), Department of Geosciences and Natural Resource Management, University of Copenhagen, Øster Voldgade 10, DK-1350 Copenhagen K, Denmark. [46]Department of Physiological Diversity, Helmholtz Centre for Environmental Research–UFZ, Deutscher Platz 5e, 04103 Leipzig, Germany. [47]Department of Ecology and Genetics, University of Oulu, Pentti Kaiteran katu 1, Linnanmaa, Oulu, Finland. [48]Swiss Federal Research Institute WSL, Zürcherstrasse 111, 8903 Birmensdorf, Switzerland. [49]CSIC, Global Ecology Unit CREAF-CSIC-UAB, 08193 Cerdanyola del Vallès Bellaterra, Catalonia, Spain. [50]CREAF, 08193 Cerdanyola del Vallès, Catalonia, Spain. [51]Cirad, UMR EcoFoG (AgroParisTech, CNRS, Inra, Univ Antilles, Univ Guyane), Campus Agronomique, 97310 Kourou, French Guiana. [52]Department of Biology, Queen's University, Biosciences Complex, 116 Barrie St., Kingston, ON K7L 3N6, Canada. [53]Biology Department, Swedish Agricultural University (SLU), SE-750 07 Uppsala, Sweden. [54]Plant Ecology and Nature Conservation Group, Wageningen University and Research, 6700 AA Wageningen, The Netherlands. [55]British Columbia Public Service, Vancouver, Canada. [56]Department of Biology, University of Saskatchewan, Saskatoon, SK S7N 5E2, Canada. [57]Department of Botany and Biodiversity Research, University of Vienna, Rennweg 14, 1030 Vienna, Austria. [58]Climate Change Science Institute and Environmental Sciences Division, Oak Ridge National Laboratory, 1 Bethel Valley Road, Oak Ridge, TN 37831-6134, USA. [59]Department of Biological Sciences and Bjerknes Centre for Climate Research, University of Bergen, N-5020 Bergen, Norway. [60]Institute of Biological and Environmental Sciences, University of Aberdeen, Aberdeen, AB24 3FX Scotland, UK. [61]Department of Biology, Vrije Universiteit Brussel (VUB), Pleinlaan 2, 1050 Elsene, Brussles, Belgium. [62]Organismal and Evolutionary Biology Research Programme, University of Helsinki, PO Box, 65FI-00014 Helsinki, Finland. [63]Département des Sciences de l'environnement et Centre d'études nordiques, Université du Québec à Trois-Rivières, 3351boul. des Forges, Québec, Canada. [64]School of Environmental Studies, University of Victoria, David Turpin Building, B243 Victoria, BC, Canada. [65]Department of Aquatic Ecology, Eawag, the Swiss Federal Institute for Aquatic Science and Technology, Überlandstrasse 133, CH-8600 Duebendorf, Switzerland. [66]Department of Biology, University of Copenhagen, Terrestrial Ecology Section, Universitetsparken 15, DK-2100 Copenhagen Ø, Denmark. [67]Research Institute for Nature and Forest (INBO), Havenlaan 88 bus 73, 1000 Brussels, Belgium. [68]Department of Bioscience, Aarhus University, Frederiksborgvej 399, 4000, Roskilde, Denmark. [69]Department of Biological Sciences, Florida International University, 11200S.W. 8th Street, Miami, FL 33199, USA. [70]Department of Ecology and Plant Geography, Moscow State Lomonosov University, 119234 Moscow1-12 Leninskie GoryRussia. [71]Department of Ecology and Evolution, University of Lausanne, Bâtiment Biophore, Quartier UNIL-Sorge, 1015 Lausanne, Switzerland. [72]Department of Natural History, NTNU University Museum, Norwegian University of Science and Technology, NO-7491 Trondheim, Norway. [73]Institute of Northern Engineering, University of Alaska, Engineering Learning and Innovation Facility (ELIF), Suite 240, 1764 Tanana Loop, Fairbanks, AK 99775-5910,

USA. [74]School of Environment, Resources and Sustainability, University of Waterloo, 200 University Avenue West, Waterloo, ON N2L 3G1, Canada. [75]Centre for Integrative Ecology, School of Life and Environmental Sciences, Deakin University, 75 Pigdons Rd, Waurn Ponds Victoria, 3216, Australia. [76]Department of Geography, University of Bonn, Meckenheimer Allee 166, D-53115 Bonn, Germany. [77]Biological and Environmental Sciences, Faculty of Natural Sciences, University of Stirling, Stirling, FK9 4LA Scotland, UK. [78]Department of Ecology, University of Innsbruck, Innrain 52, 6020 Innsbruck, Austria. [79]Environmental Change Institute, School of Geography and the Environment, University of Oxford, 3 South Parks Road, Oxford, OX1 3QY, UK. [80]Rocky Mountain Biological Laboratory, 8000 Co Rd 317, Crested Butte, CO 81224, USA. [81]Department of Environmental Science, Policy, and Management, University of California, Berkeley, CA 94706, USA. [82]Pacific Northwest National Laboratory, Joint Global Change Research Institute, 5825 University Research Ct, College Park, MD 20740, USA. [83]School of Biosciences and Veterinary Medicine-Plant Diversity and Ecosystems Management Unit, Univeristy of Camerino, Via Gentile III Da Varano, 62032 Camerino, Italy. [84]DBSV-University of Insubria, Via Dunant, 3, 21100 Varese, Italy. [85]Institute of Arctic Biology, University of Alaska Fairbanks, Fairbanks, AK 99775, USA. [86]Jonah Ventures, 1600 Range Street Suite 201, Boulder, CO 80301, USA. [87]Department of Animal Ecology and Tropical Biology, University of Würzburg, Am Hubland, 97074 Würzburg, Germany. [88]Institute for Alpine Environment, EURAC Research, Viale Druso, 1, 39100 Bolzano, Italy. [89]Department of Organismic and Evolutionary Biology, Harvard University, 52 Oxford Street, Cambridge, MA 02138, USA. [90]Institute of Systematic Botany and Ecology, Ulm University, Albert-Einstein-Allee 11, D-89081 Ulm, Germany. [91]Institute of Biology and Environmental Sciences, University of Oldenburg, Carl-von-Ossietzky-Strasse 9-11, 26129 Oldenburg, Germany. [92]Senckenberg Biodiversity and Climate Research Centre, 60325 Frankfurt, Germany. [93]Institute of Agricultural and Environmental Sciences, Estonian University of Life Sciences, Fr.R.Kreutzwaldi 1, 51006 Tartu, Estonia. [94]Graduate School of Agriculture, Kyoto University, Sakyo-ku, Kyoto 606-8502, Japan. [95]Vegetation, Forest and Landscape Ecology, Wageningen University and Research, P.O. Box 47NL-6700 AA Wageningen, The Netherlands. [96]Ecology and Conservation Biology, Institute of Plant Sciences, University of Regensburg, Regensburg, Germany. [97]Department of Forest Resources, University of Minnesota, 115 Green Hall, 1530 Cleveland Ave. N., St. Paul, MN 55108, USA. [98]Hawkesbury Institute for the Environment, Western Sydney University, Penrith, NSW 2751, Australia. [99]Department of Biology, Santa Clara University, 500 El Camino Real, Santa Clara, CA 95053, USA. [100]Department of Biology, Algoma University, 1520 Queen Street East, Sault Ste., Marie, ON P6A 2G4, Canada. [101]Komarov Botanical Institute, Professor Popova Street, 2, St Petersburg, Russia. [102]Institute for Biodiversity and Ecosystem Dynamics, University of Amsterdam, Postbus 94240, 1090 GE Amsterdam, Netherlands. ✉email: haydnjdthomas@gmail.com

