## [Peer Review File · Nature Communications]

Reviewers' comments:

Reviewer #1 (Remarks to the Author):

The paper reports on an analysis of an important new dataset of functional traits of tundra species in the context of published global patterns of plant ecological strategy quantified across all environments. As such, it represents a novel and significant addition to the literature that justifies publication in a general science journal of this stature. The statistical analysis and interpretation of results is robust and I largely agree with the headline (surprising) story that the global economic spectrum of plant strategies (acquisitive vs. conservative) is largely covered by tundra species.

However, I found myself more interested in the contrasts between the tundra dataset and the larger TRY dataset which is less well developed in the current version. An important question being addressed in the paper is the capacity for tundra communities to adapt to environmental change in terms of the within species variation in traits. While it is true that tundra trait variation covers a similar range in terms of PCA2 when compared to the global data, individuals are clearly weighted towards a more conservative strategy (bottom left hand corner of ordination) with some TTT points outside of the trait envelope delimited by the TRY data. I would like to make a few suggestions for how this aspect of the work could be better described in the paper.

1) Combine trait data collected in environments north of 60 deg. (Fig S13) with data collected in sites with annual $T < 0$ (Fig S20) into a single group that could be thought of as 'true' tundra trait data (while I acknowledge the issue with defining tundra on environmental criteria). These two groups largely tell the same story and combining them would also allow the Supp. Fig count to be reduced (currently a bit tedious to plough through lots of similar figures).

2) Include the figure of trait distributions and trade-offs for this combined group (currently Figs S15 and S22) in the main paper - while it supports the view that the trait range is similar to the global dataset it also makes the point that there is also a shift in the weighting of resource related traits in the tundra data as well as the size traits.

3) Include this third group either as an additional group in the PCA (Fig2) or in another panel comparing 'true tundra data with TRY global data (could possibly include centroids of groups on ordination?). This would help distinguish the trait space currently occupied in the tundra from the potential adaptive potential within tundra species.

4) I would like to see some discussion on the ecological processes that explain this weighting towards conservative strategies with a particular note of species that expand the trait envelope beyond that currently defined in the TRY publications. I am particularly intrigued by the three (TTT?) points in the extreme bottom left hand corner of the PCA. Do these individuals challenge current plant ecological

strategy theory in terms of evolutionary constraints and trade-offs between traits? Where do they fit in the global leaf economic spectrum? What species are they?

Finally, a comment on trees - the analysis largely excludes trees a-priori by selecting samples above the tree line (I think this needs to be made more explicit in the main text). I agree that this largely explains the empty space on the right of the ordination in the tundra. This could be made clearer by distinguishing trees in the ordination (Fig2). While I appreciate that the explanation for why trees don't grow in the tundra may not be straightforward, some further discussion of this in the context of the traits analysed here would strengthen the overall story.

Reviewer #2 (Remarks to the Author):

Summary

Thomas et al. ask whether trait covariation patterns observed globally extend to the tundra biome. They focus on 3 carbon acquisition traits and 3 size traits. They find that these traits are correlated among species the same way in the tundra as they do among species in other biomes. Second, they find that the range of size trait values is reduced in the tundra compared to the range found in the rest of the globe, while the range of carbon economic traits is not. Third, they report that most of the variation occurs among species, as opposed to within species or among functional groups. Fourth, they examine how the spatial extent of one's study affects the relative importance of intraspecific and interspecific variation, and they find that the amount of intraspecific variation increases with decreasing spatial extent.

The paper works with a unique dataset that provides extensive coverage of tundra plants. The analyses are thorough, well conducted and clearly reported. However, I also have major problems with the narrative. First, the justification of the two main questions is incorrect: two trait axes does not organize all of phenotypic diversity, and these tradeoffs have been studied in the arctic before. Second there is a number of instances where the conclusions drawn overreach beyond what the results actually show. Moreover, I don't find the results surprising or unexpected. I wasn't expecting the global trait relationships to be different in the tundra biome, and the manuscript does not provide a rationale for why we might expect them to be different. Third, a discussion of the plant size axis is necessary. from figure 2, it is not clear to me if it indeed forms an axis and if so, if it's independent from the leaf economics axis. Last, the writing is unclear in enough places that it becomes problematic.

Major comments

The setup of the two main questions is incorrect

(1) All of phenotypic diversity is NOT organized along a plane

p.1, Ins 1-2 & Ins 20-21. I disagree with the very first sentence of the abstract, which establishes the context for the study. The sentences reads "The majority of global variation in key traits determining plant fitness is thought to be explained by just two dimensions" and "Relationships among plant traits enable the diverse characteristics of global plant species to be described by just two dimensions".

p.6, Ins 19-21. Similarly, the second sentence of the introduction states: "Relationships among plant traits enable the diverse characteristics of global plant species to be described by just two dimensions"

In support of these statements, Díaz, S. et al. The global spectrum of plant form and function. *Nature* 529, 167–171 (2015), is cited. Diaz et al. find that variation in 6 key traits is mainly organized along a plane, composed of two dimensions: size and carbon economics. However, the interpretation of these results proposed in the Thomas et al manuscript, that that this plane organizes the diversity in ALL plant traits is vastly overreaching. I strongly disagree with this interpretation, because the two trait axes organize the diversity among SIX traits, not among ALL traits. The phenotype is made of thousands of traits and stating that relationships among six traits reflects all of phenotypic diversity is incorrect. There is a large number of other traits that are key to determining plant fitness, and they may well be orthogonal to the leaf economic and plant size traits. Or not. We simply don't know.

Unfortunately, this is how the manuscript justifies studying the traits they did, and the introductory narrative needs to be modified. For example, it could state that these two aspects of plant variation are important in organizing phenotypic diversity. This is very different from stating that these dimensions are the only two shaping phenotypic diversity. "Two strategy axes, resource acquisition and plant size, are important in organizing phenotypic variation" would be a true statement.

Further, these two dimensions have been known 20 years before the Diaz 2015 paper. Westoby's Leaf-Height-Seed strategy (Westoby et al. 1998. *Plant & Soil*; Westoby et al. 2002. *Ann.Rev.Ecol.Syst.*) in fact describes plant size and leaf carbon economics as two major dimensions organizing phenotypic diversity. It is important to give due credit. Interestingly, Westoby's work suggests that seed size and plant size are two different dimension, while the Diaz paper interprets their results as these two traits covarying. It would make for a richer setup to introduce and discuss the two alternative view.

(2) Trait relationships in climatic extremes HAVE been studied.

p.6 lns 6-7 “It is unknown whether global trait relationships extend to climatic extremes”.

p.6, lns 27-29. “It is thus unknown whether trait relationships are generalizable to temperature-limited biomes such as the tundra...”

This is not true. The original papers describing the relationships among leaf traits covered a range of biomes, including temperature extremes (the desert), and most importantly, including arctic tundra! See figure inset in Figure 1 of Reich et al. 1997. PNAS They had sites in both tundra and the desert. Table 1 of Reich et al. 1999. J.Ecol also shows that one site in the arctic tundra and another in the desert. The motivation for this study is thus incorrect.

I do not find the results to be particularly unexpected. Even if trait relationships in alpine and tundra systems were unstudied, or understudied, why is it surprising to find that the relationships among size and carbon economic traits hold? It would be helpful if the manuscript would discuss the reasons why one might expect those relationships to be similar or different from those observed in other biomes. There seems to be the implicit hypothesis: that we wouldn't normally expect the relationships to be present in the arctic and tundra. Why not?

I would rewrite the intro to acknowledge the work that has been done and emphasize what this expanded dataset brings beyond what Reich's original work already did.

(3) Some of the conclusions drawn are overreaching beyond what the results actually show

P10 ln 133. “our findings indicate that species-level variation comprises the majority of the global spectrum of plant form and function”. This is incorrect. Your results are for your TUNDRA plants. Stick to conclusions that your data can actually support... Delete “global” and add “in tundra plants” after “function”.

p.13 lns 172-173: “We find considerable support for environmental filtering”. Wait, what? First, this is disconnected from the introduction, as the idea of environmental filtering was not brought up at all. Second, you didn't even discuss or measure environmental gradients. Nor environmental filtering for that matter anywhere in the intro, methods or results. Third, You haven't SHOWN environmental filtering. At best, your results are consistent with environmental filtering. Other mechanisms can also lead to a different set of traits in this biome than from the global range, including but not limited to drift, dispersal limitation and phylogenetic history. It needs to be crystal clear that you are speculating that the resulting trait space is likely due to the different climate which selects for a different set of traits. Don't confuse speculation and demonstration.

P13 lns 177-178. Sentence starting with “In the tundra, where...”. Qualify this statement. A little bit of modesty/being careful wouldn't hurt. For example, you could write “If the spatial patterns

observed in our study are indicative of temporal response” Space-for-time equivalence is a huge assumption.

P13 Ins 181-182. “community trait change may require immigration of species from warmer sites. You are making the implicit assumption that the community trait mean will successfully track the new climate. What if species don't migrate fast enough, and you just end up with a shorter mean community height than you had predicted? State your assumption by qualifying the sentence. For example: "In order for community mean height to track the warming climate, ..."

P13 Ins 182-184. “The broad range of resource economic traits found within and across tundra plant communities could additionally enhance resilience in the face of climate change” I don't see the logic behind this sentence because we don't see an association between resource economic traits and climate. The fact that your results find all values of resource-economic traits within the tundra, as well as within other biomes, in fact suggests that resource economic traits are not associated with climate, right? So how can a broad range of values in these traits enhance resilience to climate change, if they are not associated with climate? Unless I am missing some important result.

P14 In 199. “relationships and trade-offs among fundamental plant traits are generalizable across lineages and among biomes...”. Careful with over-generalization I don't think you have shown that they hold across ALL lineages and ALL biomes. What about desert plant, where most photosynthetic tissue is on the bark/stem? Your study shows the relationships are generalizable to the arctic lineages and biome. Which I think we already knew, as stated in comment #2.

(4) I am unclear on the interpretation of the plant size axis

P9 In 101. I am unclear about how the size dimension is interpreted from the results.

First, from looking at trait clustering in the PCA, it is unclear to me that the 3 size traits form a single entity because leaf area is not any closer to the other size traits than to the leaf economic traits. Leaf Area is positioned half-way between Leaf N (an economic trait) and Leaf Height (a size trait) in both panels of Figure 2. So although leaf area was classified a priori as a size trait, is it also related to the LES? It appears correlate as much or more with PC1 as with PC2.

Second, from a Principal Component standpoint, the 3 size traits seem to be (on average) at 45 degrees between PC1 and PC2. So are we not finding that the size traits are also partly correlated with leaf economic traits?

The authors need to run a significance test with the `ordiellipse` function of the `BiodiversityR` package to test for significant loading of each variable on each axis to show which traits load significantly on which PCs. This will affect the interpretation of the results.

(5) Lack of clarity in writing.

This could be a minor comment, but given the frequency of instances when clarifications are needed, it significantly takes away from the ability to understand the manuscript. Here are the sentences that need clarification:

5.1- P6 ln7 "... and if relationships are confounded by trait variation within species". I don't know what that statement means. What relationships? Interspecific? Global? Do you mean if the interspecific relationships are affected by the intraspecific variation?

5.2- P6 lns 10-11 "...Tundra plants demonstrated remarkable consistency in resource economic traits, but not size traits". I am not sure what "consistency" means in this context. The same relationships as those found in other biomes? I am also unclear what is happening with size traits. What is not consistent about them? If trait relationship in tundra plants are "not consistent" with those reported in global distributions, how can they exhibit the same two dimensions of trait variation?

5.3 - P6 lns 12-13 "Three quarters of trait variation occurred among species, mirroring global estimates". Estimates of what? Of the extent of interspecific variation?

5.4 - P6 lns 12-13 "Except at local scales" This is a confusing bit of information without further background on the question and methods. Does the importance of interspecific variation depend on the scale? If so, what do you mean by local? Save that result for somewhere where you have the space to explain what you mean. In the abstract, it is just confusing.

5.5 - P6 lns 14-15 "informing prediction of plant community change in a warming world". But the world is warming and all biomes will get warmer. So why is understanding relationships in the coldest biomes informative when all the biomes will get warmer? The world will include fewer arctic climates. The statement would be true if arctic biome were to become more common or colder, but it's the opposite.

5.6 - P6 ln 34. "within species variation can invalidate trait relationships". This sentence is confusing. Specify which trait relationships you are talking about. Global interspecific? Also, what does "invalidate" mean? The relationships might be absent or reversed?

5.7 - P7 lns 37-38: "within-species variation [...] has been hypothesized to be greater for species that span large biogeographical gradients or in extreme environments and at small geographical scales." I

can guess why that would be the case, but I shouldn't have to be guessing. Please explain the rationale behind those arguments. The first and last scenario appear to be contradictory. Small spatial scales?

For large biogeographical range, is it because of the large niche breadth (mechanism is adaptation to a wide variety of niches), or is it because of the species having many populations whose geographical isolation increase their ability to drift in different directions?

5.8 - P7 In 39 replace "location to challenge" with "system to test"

5.9 - P7 In 53 "... yet will be constrained". What will be constrained? The range of trait values ?

Why this hypothesis? Provide a rationale for why would expect this hypothesis a priori. Is this like a null hypothesis?

Currently, it appears as if it as added a posteriori to the intro after finding this result. If we have no reason to hold this hypothesis, then just say that you are asking the question of how trait relationships differ in the tundra (although that has been answered in the two papers cited earlier) and state that we don't have expectations.

5.10 - P7 In56. Replace "relative to" with "than at"

5.11 - P7 In 59 "To examine constraints to trait expression". Awkward wording. What does that mean? To examine trait ranges?

5.12 - P7 Ins 66-67 "We tested whether sources of trait variation were dependent upon spatial scale". I am not sure what you mean here. You mean you explored how your variance partitioning results change as your spatial scale increases?

5.13- P9 Ins 92-93 ""This broad range of resource economic trait values occurred independently of the bioclimatic range of tundra species". I don't understand what that means. I read this as meaning that species with narrow or broad niche ranges all expressed a broad range of resource economic trait values? If that's what you mean, I have a hard time believing it.

5.14- P9 In 100. Replace "partially constrained trait expression" with "showing a limited range of trait values"

5.15- P9 In 104. Add "relative to the global data" after "was reversed"

5.16- p10 ln 123. Replace “principles” with “assumptions”

5.17- p10 ln 125. Replace “estimates” with “means”

5.18- p10 ln 126. “large species ranges in the tundra (Fig 3b)”. Do tundra species have larger environmental niche or geographic ranges (spatial extent covered) than species in other biomes?

I don't see how Figure 3b illustrates this. It shows the coefficient of variation for each trait (how variable trait values are relative to one another).

5.19- P10 ln 127. Replace “varied substantially by traits” with “changed substantially among traits”. “variation varied” is confusing.

5.20- P10 ln 129. Is 25.6 the average contribution of functional group to total variance across all 6 traits?

Why do you call 25% "little"? It seems to be as much as species-level variation for LMA and LDMC.

5.21- P10 ln 130 delete “biogeographic”

5.22- P10 lns 132-133. “species-level variation comprises the majority of the global spectrum...”. This is only true for all your traits if you consider that the total species-level variation is the among-functional group + within-functional group species differences.

If you use a strict definition of species-level, (as I think you used in your analyses - i.e. within functional group – this circles back to my previous comment asking for this clarification), then this statement is only true for 3/6 of your traits.

I encourage the authors to change the legend and ms terminology to specify that "among species" refers to "among-species within functional groups".

5.23- P10 ln 133. Delete “global” and add “in tundra plants” after “function”. Your results are for your TUNDRA plants. Stick to conclusions that your data can actually support...

5.24- P10-11 lns 133-135. “underlining the importance of species richness and turnover in determining plant community characteristics, trait diversity, and linkages to ecosystem function.”. This statement goes in the discussion section, not the results section.

5.25- P11 In151. Delete “clear scaling” (we don’t know what you mean by scaling), add “is the first to” after ‘study’, and add ‘between spatial scale and extent of intraspecific variation’ after “relationship”.

5.26 – P12 In 156. This is true for 3 of the 4 traits, but not for N. You need to specify that to be correct.

5.27 - P13 Ins 167 “including resource acquisition, survival, competition and reproduction.” This is a very broad claim. It would help if you reminded us of which trait is associated with which of these properties. E.g. “resource acquisition (LMA,LNC,LDMC), survival (?), competition (plant height) and reproduction (seed size).

5.28 - P13 In 168 “conceptualization”. I don’t know what that means in the context of this sentence. The measurement of? I think the sentence would be clearer with no loss of information if you just delete this bit and write “..., trait space should include seldomly measured traits...”

5.29 - P13 In 169. “underrepresented traits”. do you just mean traits you haven't measured in this study? Or do you mean traits that are never measured? It would be more informative if you were more specific than simply stating that we need to measure more traits. Then the comment “that capture critical links to ecosystem function”. You are opening a bag of worms here. Are you saying your traits have poor link to ecosystem function? That was part of the justification for measuring them earlier... Also, do we know that below-ground traits have stronger links to ecosystem function? Which traits? which function? This sentence is so vague that it does not bring any value. Either delete it or state something more specific.

Other Minor comments

P7 In 47 “89% of the tundra species pool”. That's great coverage. What kind of replication do you get for each species? Multiple populations? Multiple individuals? TRY is a bit infamous for reporting trait means.

P7 Ins 63-66 Are species nested within each functional group, or does the species scale go across functional groups? in the later case?

P6 In 67 replace “were dependent upon” with ‘depend on the’. Using active verbs is always clearer.

P8 Figure 1 panel C. replace “Observations” in y-axis label of upper panel with “Trait observations”

P9 In 94. Figure S1 seems to contradict Figure 2B, where the trait hypervolume occupied by cold and warm species is smaller than the hypervolume occupied by mid-temp species. Imagine drawing convex hulls around each of these 3 species group. The mid-temp species clearly would cover a much larger area than the warm and cold species. Where am I going wrong and how can we avoid such confusion on the part of the reader?

P9 Ins 95-97. "tundra, this variation in plant leaf resource economics is remarkably high, and suggests that plants have developed a wide range of ecological strategies to cope with extreme conditions" . To me, this is the that's the most interesting finding. I would write the paper around that result instead of around asking whether trait relationships extend to the arctic, which we know they did from Reich's work.

P9 Ins108-110. Move this statement to the discussion section. This is an interpretation, not a result.

P13 In 175 In line with my previous comments, replace "that the two major axes" with "that two of the major axes"

P13 In 186. Replace "inform" with "support".

P14 In195. "better incorporate hierarchical trait variation". This statement makes me wonder how hierarchical trait variation is currently specified in those models, and how you propose to improve it.

30th August 2019

We thank the reviewers for their comments and feedback on our manuscript, 'Global plant trait relationships extend to the climatic extremes of the tundra biome'. We have now completed our revisions and hope that we have been able to address all comments and concerns. We have made the following major changes, in addition to multiple smaller improvements throughout the text:

1. Rewording and reorganisation of the abstract and introduction to:
 - Better acknowledge the range of work conducted in this field to date
 - More clearly state that the axes tested in this study apply to six key traits, and not to all potential plant traits or life strategies
 - Expand upon and better justify our hypotheses
2. Greater exploration of the differences found between global and tundra datasets, including differences in the contribution of traits to the two axes of variation (plant size and resource acquisition).
3. Re-analysis of trait variation within the tundra ('temperature classifications').
4. Combination of 'extreme' tundra analyses into one dataset.
5. Clarification of statements and implications throughout the manuscript.

All changes are outlined in detailed point-by-point responses to reviewer comments below.

Yours sincerely,

Haydn Thomas
On behalf of all authors.

Reviewers' comments:

Reviewer #1 (Remarks to the Author):

However, I found myself more interested in the contrasts between the tundra dataset and the larger TRY dataset which is less well developed in the current version. An important question being addressed in the paper is the capacity for tundra communities to adapt to environmental change in terms of the within species variation in traits. While it is true that tundra trait variation covers a similar range in terms of PCA2 when compared to the global data, individuals are clearly weighted towards a more conservative strategy (bottom left hand corner of ordination) with some TTT points outside of the trait envelope delimited by the TRY data. I would like to make a few suggestions for how this aspect of the work could be better described in the paper.

1) Combine trait data collected in environments north of 60 deg. (Fig S13) with data collected in sites with annual $T < 0$ (Fig S20) into a single group that could be thought of as 'true' tundra trait data (while I acknowledge the issue with defining tundra on environmental criteria). These two groups largely tell the same story and combining them would also allow the Supp. Fig count to be reduced (currently a bit tedious to plough through lots of similar figures).

We have combined the two supplementary analyses as suggested. This now results in a single subset of trait records that are found either north of the Arctic circle (66.5°N ; note this revised cut-off from 60°N in the previous manuscript) or at sites with a mean annual temperature below 0°C .

2) Include the figure of trait distributions and trade-offs for this combined group (currently Figs S15 and S22) in the main paper - while it supports the view that the trait range is similar to the global dataset it also makes the point that there is also a shift in the weighting of resource related traits in the tundra data as well as the size traits.

We suggest that to avoid repetition and potential confusion in the main text we use the full tundra dataset only in this figure, as per the original text, and simplify the supplementary figures using these two combined groups. However, if the reviewer and editor consider it essential, we would also be happy to include two new panels within Figure 2 to more clearly demonstrate the shift in the weighting of both resource and size related traits within tundra data:

Figure 2: Global trait relationships are maintained in the tundra biome despite constrained size, but not resource economic, traits among tundra species. (a) Global trait-space defined by six plant traits for 1,358 plant species in the global dataset (grey points) and 219 tundra species (blue points). (b) Distribution of trait

space for tundra species only. Points are coloured by temperature category, corresponding to the mean annual temperature of trait collection sites for each species (Cold < -1°C, Mid >-1 °C but < 1°C, Warm > 1°C, Fig. S1). (c) Global trait space and (d) tundra trait space using only species found north of the Arctic Circle (66°N) or at sites with a mean annual temperature >0°C. Arrows indicate the direction and weighting of trait vectors.

3) Include this third group either as an additional group in the PCA (Fig2) or in another panel comparing 'true tundra data with TRY global data (could possibly include centroids of groups on ordination?). This would help distinguish the trait space currently occupied in the tundra from the potential adaptive potential within tundra species.

We believe that this is covered by a revised analysis in Figure 2b, which separates species based on the mean temperature of trait collection sites for each species. We argue that this approach more accurately represents a 'thermal niche' of each species (rather than the more straightforward presence-absence approach taken in the reduced 'true' tundra dataset), and so is more appropriate for addressing the question posed here. In addition, we now include further discussion on the adaptive potential within tundra species (L104-107 & L199-205).

"Species with faster resource-related traits and larger size-related traits were also associated with warmer environments within the tundra (Fig S1), potentially informing the adaptive capacity to climate warming within and among tundra plant communities."

"If the spatial patterns observed in our study are indicative of temporal responses (Elmendorf et al., 2015; Bjorkman et al., 2018), climate change will likely shift trait distributions towards increased plant height, leaf area and seed mass, as has already been observed at some sites (Hudson et al., 2011; Myers-Smith et al., 2019a), and for plant height at the biome scale (Bjorkman et al., 2018). However, since the majority of trait variation occurs among species, shifts in community-level traits may require immigration of species from warmer sites, and may thus occur more slowly than would be predicted from biogeographic gradients (Elmendorf et al., 2015; Bjorkman et al., 2018)."

4) I would like to see some discussion on the ecological processes that explain this weighting towards conservative strategies with a particular note of species that expand the trait envelop beyond that currently defined in the TRY publications. I am particularly intrigued by the three (TTT?) points in the extreme bottom left hand corner of the PCA. Do these individuals challenge current plant ecological strategy theory in terms of evolutionary constraints and trade-offs between traits? Where do they fit in the global leaf economic spectrum? What species are they?

We now include additional discussion of the position of tundra species within global trait space (L93-96).

"Many tundra species, such as the prostrate, small-leaved, and wind-dispersed evergreen shrub *Cassiope hypnoides* were located at the very edge of global trait space, consistent with adaptation to the extreme environmental conditions in the tundra (Billings, 1987)"

Finally, a comment on trees - the analysis largely excludes trees a-priori by selecting samples above the tree line (I think this needs to be made more explicit in the main text). I agree that this largely explains the empty space on the right of the ordination in the tundra. This could be made clearer by distinguishing trees in the ordination (Fig2). While I appreciate that the explanation for why trees don't grow in the tundra may not be straightforward, some further discussion of this in the context of the traits analysed here would strengthen the overall story.

This is a very interesting question.

First, we agree that understanding trait differences at treeline is itself of interest - not only due to the qualitative difference in ecosystem diversity, structure and function, but also in informing potential trait

change as the climate warms. Indeed, some tree species are occasionally found in our tundra plots - either in dwarf form or (more often) as seedlings. For the purpose of this analysis we discussed how best to include these species, given that the vast majority of their trait records are taken outside of tundra plots, and that the coordinates of trait collection sites are not always known. For this particular analysis, we decided that tree species (i.e. those over 5m in height) should not be included as tundra species - as the absence of trees is one of the defining features of the tundra biome. We chose to exclude species in this way so as not to filter trait values within species in any way, which could invalidate our analyses of trait variation.

Second, we agree that the distinction between trees and non-trees is certainly an interesting feature of the global trait dataset. This distinction is explored in some detail in Diaz et al's (2016) paper (Figure R1 below); we do not feel we would be well placed to add to Diaz's analysis in this manuscript.

Figure R1: Woody and non-woody species comprise distinct group in global trait-space. From (Díaz et al., 2016).

Nevertheless, we have plotted the position of common boreal trees, or those removed from the tundra dataset (*Alnus incana*, *Alnus viridis*, *Betula pendula*, *Picea orientalis*, *Pinus sylvestris*, *Populus tremula*, *Sorbus aucuparia*) within global trait space alongside tundra species. This illustrates that boreal tree species form a distinctly different group to tundra species, as would be predicted from Diaz et al (2016). Note this only includes species for which we have complete trait data within this tundra dataset. We would be happy to include this as a supplementary figure if the reviewer thinks this would be helpful.

Finally, we have made it clearer within the manuscript that trees are not included in this tundra dataset since part of the selection process for tundra species is that they would be those above treeline (L96-101).

"Given that tundra communities are found above treeline, and therefore by definition exclude tree species, we expected to see reduced plant height among tundra species compared to global species. However, lower plant height also corresponded with smaller leaf area and seed mass (Fig. 2a, axis 1, Fig. S1), as would be predicted from global trait relationships."

We have also slightly expanded our discussion of trees (and plant height) (L120-123) to make interpretation clearer:

"Nevertheless, trait co-variation was maintained in the tundra despite the absence of trees, which comprise half of global trait space (Díaz et al., 2016) and have been a focus of many previous studies of plant trait relationships (Westoby, 1998; Wright et al., 2004; Westoby and Wright, 2006; Chave et al., 2009)."

Reviewer #2 (Remarks to the Author):

Summary

The paper works with a unique dataset that provides extensive coverage of tundra plants. The analyses are thorough, well conducted and clearly reported. However, I also have major problems with the narrative. First, the justification of the two main questions is incorrect: two trait axes does not organize all of phenotypic diversity, and these tradeoffs have been studied in the arctic before. Second there is a number of instances where the conclusions drawn overreach beyond what the results actually show. Moreover, I don't find the results surprising or unexpected. I wasn't expecting the global trait relationships to be different in the tundra biome, and the manuscript does not provide a rationale for why we might expect them to be different. Third, a discussion of the plant size axis is necessary. From figure 2, it is not clear to me if it indeed forms an axis and if so, if it's independent from the leaf economics axis. Last, the writing is unclear in enough places that it becomes problematic.

Major comments

The setup of the two main questions is incorrect

(1) All of phenotypic diversity is NOT organized along a plane

p.1, lns 1-2 & lns 20-21. I disagree with the very first sentence of the abstract, which establishes the context for the study. The sentences reads "The majority of global variation in key traits determining plant fitness is thought to be explained by just two dimensions" and "Relationships among plant traits enable the diverse characteristics of global plant species to be described by just two dimensions".

p.6, lns 19-21. Similarly, the second sentence of the introduction states: "Relationships among plant traits enable the diverse characteristics of global plant species to be described by just two dimensions"

In support of these statements, Díaz, S. et al. The global spectrum of plant form and function. *Nature* 529, 167–171 (2015), is cited. Diaz et al. find that variation in 6 key traits is mainly organized along a plane, composed of two dimensions: size and carbon economics. However, the interpretation of these results proposed in the Thomas et al manuscript, that that this plane organizes the diversity in ALL plant traits is vastly overreaching. I strongly disagree with this interpretation, because the two trait axes organize the diversity among SIX traits, not among ALL traits. The phenotype is made of thousands of traits and stating that relationships among six traits reflects all of phenotypic diversity is incorrect. There is a large number of other traits that are key to determining plant fitness, and they may well be orthogonal to the leaf economic and plant size traits. Or not. We simply don't know.

Unfortunately, this is how the manuscript justifies studying the traits they did, and the introductory narrative needs to be modified. For example, it could state that these two aspects of plant variation are important in organizing phenotypic diversity. This is very different from stating that these dimensions are the only two shaping phenotypic diversity. "Two strategy axes, resource acquisition and plant size, are important in organizing phenotypic variation" would be a true statement.

Further, these two dimensions have been known 20 years before the Diaz 2015 paper. Westoby's Leaf-Height-Seed strategy (Westoby et al. 1998. *Plant & Soil*; Westoby et al. 2002. *Ann.Rev.Ecol.Syst.*) in facts describes plant size and leaf carbon economics as two major dimensions organizing phenotypic diversity. It is important to give due credit. Interestingly, Westoby's work suggests that seed size and plant size are two different dimension, while the Diaz paper interprets their results as these two traits covarying. It would make for a richer setup to introduce and discuss the two alternative view.

We agree. We have rephrased the abstract and introduction to more clearly and accurately reflect what can be described by these two dimensions of plant form and function.

We have also reworded the introductory text to indicate that there are indeed important precedents to Diaz et al.'s work, and included additional references as appropriate.

Abstract (L2-4):

“The majority of variation in six major plant traits critical to the growth, survival and reproduction of global plant species is thought to be organised along just two dimensions, corresponding to strategies of plant size and resource acquisition.”

Introduction (L20-28):

“Despite the vast diversity of life on Earth, vascular plants are limited by trade-offs in leaf (Wright et al., 2004), wood (Chave et al., 2009), seed (Westoby et al., 1992) and root (Iversen et al., 2015) traits, enabling the characteristics of global plant species to be organised along a few general dimensions (Westoby, 1998; Westoby and Wright, 2006; Reich, 2014; Díaz et al., 2016). Two dimensions, plant size (large and woody vs. small and non-woody) and resource economics (acquisitive vs. conservative), have been shown to describe the majority of variation in six widely-sampled plant traits, which together represent key differences in plant form and function (Díaz et al., 2016). Such trait relationships predict community assembly (McGill et al., 2006; Cornwell and Ackerly, 2009) and ecosystem functions (Lavorel and Garnier, 2002; Suding et al., 2008) across biogeographic gradients (Bjorkman et al., 2018) and in response to environmental change (Lavorel and Garnier, 2002; Moran et al., 2016).”

We do not believe that this alters the justification for either the study or the selection of these six traits, since they underpin the two dimensions of global trait space tested here, and are the most widely sampled plant traits among global plant species.

(2) Trait relationships in climatic extremes HAVE been studied.

p.6 Ins 6-7 “It is unknown whether global trait relationships extend to climatic extremes”.

p.6, Ins 27-29. “It is thus unknown whether trait relationships are generalizable to temperature-limited biomes such as the tundra...”

This is not true. The original papers describing the relationships among leaf traits covered a range of biomes, including temperature extremes (the desert), and most importantly, including arctic tundra! See figure inset in Figure 1 of Reich et al. 1997. PNAS They had sites in both tundra and the desert. Table 1 of Reich et al. 1999. J.Ecol also shows that one site in the arctic tundra and another in the desert. The motivation for this study is thus incorrect.

We agree that we could better incorporate the work of Reich et al. (1997, 1999), and this is now reflected in a revised introduction to the text. However, we do not agree that these two studies clearly demonstrate that the trait relationships examined by this study extend to the tundra, for three main reasons:

First, the relationships explored by Reich et al. relate only to the leaf economic spectrum - examining SLA, leaf nitrogen, leaf lifespan, and photosynthetic rate. Although these relationships have been explored at some tundra sites, the whole-plant trade-offs identified by Diaz et al. (and others) have not. Part of the motivation for this study was to examine whether these whole-plant relationships were upheld in the tundra, particularly given that certain size-related traits (such as plant height) are so dramatically constrained within the tundra biome.

Second, it is not correct to say that Reich's papers included sites from the Arctic tundra. The tundra site included in his studies is located in the alpine tundra in Colorado (Niwot Ridge). We do not by any means discount this as a tundra site - indeed there is an International Tundra Experiment site at this location. However, we disagree that the trait relationships found at this single alpine site would necessarily be representative of the Arctic and alpine tundra biome. Further to this, Reich's analyses includes three tree

species, which have been the focus of many trait-relationship studies, yet are by definition excluded from our study of tundra species.

Third, a key motivation of this study was to explore relationships within and across the tundra biome, incorporating species at the cold extreme of plant life, and investigating the extent and sources of trait variation across this large and variable biome. We do not argue that trait data or studies are not available for the tundra biome (e.g. see Fig 1), but rather that tundra data are largely in the minority. For example, the tundra has featured in < 3% of plant trait research in the last decade and comprises < 5% of data in TRY (Myers-Smith et al., 2019b). This becomes important if key differences between biomes are obscured once data are compiled for all global plants, as is the case for disturbance driven systems (e.g. see Wigley et al., 2016). The reviewer is right to highlight that Reich et al.'s 1997 and 1997 papers are some of the few to explicitly test for differences in trait relationships among biomes, and we now reflect this more accurately within the text. However, we believe that our study is uniquely placed to test this question at the full biome-scale - covering 219 tundra plant species across 1,400 unique locations and over 50,000 trait records (compared to seven species at one alpine tundra site in the Reich et al study). Thus, although our results for the leaf economic spectrum (for example, see Fig S2) do align with Reich's findings, we believe that we are able to demonstrate this in a highly robust manner, as well as providing additional insight into trade-offs at the whole-plant level, and trait variation within the tundra biome as a whole.

In light of these comments, we have now more clearly identified throughout the text how our study differs from previous studies, and more clearly acknowledge the important contribution that previous studies have made to establishing the trait relationships we are testing here, both in the tundra and elsewhere.

I do not find the results to be particularly unexpected. Even if trait relationships in alpine and tundra systems were unstudied, or understudied, why is it surprising to find that the relationships among size and carbon economic traits hold? It would be helpful if the manuscript would discuss the reasons why one might expect those relationships to be similar or different from those observed in other biomes. There seems to be the implicit hypothesis: that we wouldn't normally expect the relationships to be present in the arctic and tundra. Why not?

We agree that not all will find this particular result - that trait relationships are consistent in the tundra - unexpected. As is the case when testing any hypothesis, our camps were split: around half of the co-authors were not surprised, the other half were (as was Reviewer 1). We believe that the analyses presented here provide good evidence to support the reviewer's expectation that trait relationships (especially for the LES) should be universal. However, we also argue that a significant proportion of both the existing literature (e.g. see Fajardo and Piper, 2011; Shipley et al., 2016; Wigley et al., 2016; Messier et al., 2017), and the scientific community who encouraged us to conduct this test, did not hold this expectation.

There are several ways that trait relationships in alpine and tundra systems may differ from global patterns, particularly at the whole-plant level. Trait relationships are not always found consistently - as has found to be the case in disturbance-driven systems (Wigley et al., 2016). Tundra trait distributions are thought to be largely determined by climatic conditions and are associated with small plant size and conservative economic strategies (Molau, 1993), yet short growing seasons also drive high relative growth rates (Chapin, 1987) and leaf nitrogen (N) concentrations (Körner, 1989). As such, tundra plants may thus exhibit unique trait relationships resulting from adaptation to extreme environmental conditions. Similarly, the lack of certain functional groups may also alter relationships - most notably trees, which form a key component of many previous studies (Westoby, 1998; Wright et al., 2004; Westoby and Wright, 2006; Chave et al., 2009), and largely account for differences in plant strategy in Diaz et al.'s (2016) study.

We have now stated more clearly how extreme conditions may alter trait relationships (L30-34). We also now provide references to studies that discuss this question in more detail, and can provide some additional narrative if the editor thinks this would be helpful.

“Although some site-specific studies exist (Reich et al., 1997; Freschet et al., 2010), whole-plant trait

relationships have not been widely tested at the environmental extremes of plant life such as the cold tundra biome, where plants could exhibit rare or unique trait relationships resulting from adaptation to extreme environmental conditions (Wigley et al., 2016; Myers-Smith et al., 2019b)”

(3) Some of the conclusions drawn are overreaching beyond what the results actually show

P10 In 133. “our findings indicate that species-level variation comprises the majority of the global spectrum of plant form and function”. This is incorrect. Your results are for your TUNDRA plants. Stick to conclusions that your data can actually support... Delete “global” and add “in tundra plants” after “function”.

We have reworded this sentence to indicate that our results are consistent with the above hypothesis, rather than suggesting they will hold true for all global plants:

“Overall, our findings support the hypothesis that species-level variation comprises the majority of the global spectrum of plant form and function (Díaz et al., 2016; Shipley et al., 2016)”

p.13 Ins 172-173: “We find considerable support for environmental filtering”. Wait, what? First, this is disconnected from the introduction, as the idea of environmental filtering was not brought up at all. Second, you didn't even discuss or measure environmental gradients. Nor environmental filtering for that matter anywhere in the intro, methods or results. Third, You haven't SHOWN environmental filtering. At best, your results are consistent with environmental filtering. Other mechanisms can also lead to a different set of traits in this biome than from the global range, including but not limited to drift, dispersal limitation and phylogenetic history. It needs to be crystal clear that you are speculating that the resulting trait space is likely due to the different climate which selects for a different set of traits. Don't confuse speculation and demonstration.

We have now made clearer reference to environmental filtering in the introduction (L40-44), such that this idea is introduced earlier on, rather than appearing suddenly in the discussion.

“Within-species variation accounts for approximately 25% of trait variation at the global scale (Siefert et al., 2015), but has been hypothesised to be greater in extreme environments due to environmental filtering of trait expression (Fajardo and Piper, 2011) at small geographical scales (Albert et al., 2011; Messier et al., 2017) where species richness is low, and for species that span large biogeographical gradients (Siefert et al., 2015) due to large niche breadth.”

And (L57-60):

“We tested three hypotheses: 1) Trait expression among tundra species will be constrained relative to global trait space due to extreme environmental conditions, yet will exhibit the same two dimensions of plant form and function.”

We have also restructured the discussion paragraph in question (L191-199) to put greater emphasis on the variation in resource traits, and to emphasise that trends are consistent with environmental filtering, rather than indicative of environmental filtering.

“Tundra plant species showed remarkable variation in resource economic traits within the tundra biome relative to global trait space (Díaz et al., 2016). Given the low vascular plant diversity associated with many tundra environments, this variation in plant leaf resource economics is notably high, and suggests that tundra species have developed a wide range of ecological strategies to cope with extreme conditions and limiting resources. In contrast, tundra plant species occupied half the global range of size-related traits, potentially indicating that two of the major axes of global trait variation may be differentially selected by environmental conditions, and could thus respond differently to environmental change.”

P13 Ins 177-178. Sentence starting with “In the tundra, where...”. Qualify this statement. A little bit of modesty/being careful wouldn't hurt. For example, you could write “If the spatial patterns observed in our study are indicative of temporal response” Space-for-time equivalence is a huge assumption.

We concur that it is worth qualifying this assumption clearly. It is also worth noting that this study is not intended to present an analysis of trait *change* in the tundra, either over spatial gradients or over time, since this has already been the focus of numerous site-level studies and several biome-wide pieces of work. For example, Elmendorf et al., (2015) conduct a meta-analysis of space-for-time substitution, experimental warming, and ambient change studies across the tundra, finding that the effects of warming on community and trait change are consistent across methods. Similarly, Bjorkman et al., (2018) show that there are strong biogeographical trends in the tundra across all the traits examined in this study, but that only plant height has changed significantly over time at the biome-scale, though Steinbauer et al., (2018) find that change over time has been occurring across a greater suite of traits in alpine tundra.

Rather, one of the aims of this study was to provide a stronger understanding of some of the mechanisms underpinning these trends - for example whether trait trade-offs could contain community-level change among tundra species, and how the location of trait variation (within - vs among-species variation) could inform rates of trait change (both over space and over time).

We have now updated the text to more clearly qualify this statement (L199-202; see also the following comment):

“If the spatial patterns observed in our study are indicative of temporal responses (Elmendorf et al., 2015; Bjorkman et al., 2018), climate change will likely shift trait distributions towards increased plant height, leaf area and seed mass, as has already been observed at some sites (Hudson et al., 2011; Myers-Smith et al., 2019a), and for plant height at the biome scale (Bjorkman et al., 2018).”

P13 Ins 181-182. “community trait change may require immigration of species from warmer sites. You are making the implicit assumption that the community trait mean will successfully track the new climate. What if species don't migrate fast enough, and you just end up with a shorter mean community height than you had predicted? State your assumption by qualifying the sentence. For example: “In order for community mean height to track the warming climate, ...”

We have now expanded this sentence (L202-205) to more clearly state assumptions and further draw out the implications for space-time substitution:

“However, since the majority of trait variation occurs among species, shifts in community-level traits may require immigration of species from warmer sites, and thus occur more slowly than would be predicted from biogeographic gradients”

P13 Ins 182-184. “The broad range of resource economic traits found within and across tundra plant communities could additionally enhance resilience in the face of climate change” I don't see the logic behind this sentence because we don't see an association between resource economic traits and climate. The fact that your results find all values of resource-economic traits within the tundra, as well as within other biomes, in fact suggests that resource economic traits are not associated with climate, right? So how can a broad range of values in these traits enhance resilience to climate change, if they are not associated with climate? Unless I am missing some important result.

We have removed this line.

P14 In 199. “relationships and trade-offs among fundamental plant traits are generalizable across lineages and among biomes...”. Careful with over-generalization I don't think you have shown that they hold across ALL lineages and ALL biomes. What about desert plant, where most photosynthetic tissue is on the bark/stem? Your study shows the relationships are generalizable to the arctic lineages and biome. Which I think we already knew, as stated in comment #2.

We agree that it is important not to overgeneralise with this statement, particularly as your question regarding whether we would find the same thing in e.g. desert systems is one we have discussed ourselves in quite some depth - and would indeed make a useful study. We have therefore altered and expanded this statement to:

“Our findings demonstrate that relationships and trade-offs among six fundamental plant traits are generalisable across lineages and among biomes to the cold extremes of the planet. It remains to be tested whether such relationships are consistent at other environmental extremes, such as desert ecosystems.”

(4) I am unclear on the interpretation of the plant size axis

P9 In 101. I am unclear about how the size dimension is interpreted from the results.

First, from looking at trait clustering in the PCA, it is unclear to me that the 3 size traits form a single entity because leaf area is not any closer to the other size traits than to the leaf economic traits. Leaf Area is positioned half-way between Leaf N (an economic trait) and Leaf Height (a size trait) in both panels of Figure 2. So although leaf area was classified a priori as a size trait, is it also related to the LES? It appears correlate as much or more with PC1 as with PC2.

Second, from a Principal Component standpoint, the 3 size traits seem to be (on average) at 45 degrees between PC1 and PC2. So are we not finding that the size traits are also partly correlated with leaf economic traits?

The authors need to run a significance test with the `ordiellipse` function of the `BiodiversityR` package to test for significant loading of each variable on each axis to show which traits load significantly on which PCs. This will affect the interpretation of the results.

This is an excellent point, and we have now included an additional test of the contribution of trait loadings to each PCA axis, using a similar approach to `ordiellipse` as the reviewer suggests (see below). We find a resource-size axis structure in the tundra, with resource traits (LMA, LDMC, LeafN) significantly contributing to PCA axis 1, and size traits (height, seed mass, leaf area) significantly contributing to PCA axis 2 (Figure S4). However, as the reviewer notes, leaf area also makes a significant contribution to PCA axis 1, and as such this should be recognised in the discussion.

Comparing this to global data, we find similar direction, strength and clustering of trait loadings. However, we do not find exactly the same contributions to the two PCA axes, with leaf area associated with ‘resource traits’ and LDMC associated with ‘size traits’. This more accurately reflects the interpretations provided by Diaz et al., who emphasise the importance of stem density for plant structure, though we note that we have a more limited set of non-tundra species (due to data permissions and availability) and also use LDMC as a substitute for stem density in this study. We therefore feel it is important to draw out some of the implications of this difference, notably the importance of leaf area as a resource trait, and the importance of LDMC / stem density for plant structure.

To reflect these points we have made the following changes to the manuscript

1) We have added a section in the results which sets out these differences (L113-120):

“However, the relative importance of PCA axes was reversed relative to global data (Fig. S3), suggesting that tundra plant strategies are primarily differentiated by resource economics. Leaf area was more strongly associated with the size axis among tundra species, but with resource economics among global species (Fig S4). In contrast, leaf dry matter content was more strongly associated with resource economics in the tundra. LDMC correlates closely with stem density and is associated with plant size and structure among global plant species (Díaz et al., 2016), especially among tree species (Chave et al., 2009).”

2) We have added an additional supplementary figure (Figure S4) which indicates the contribution of trait

loadings to each PCA axis, and have also expanded the supplementary methods section as appropriate.

“Finally, we calculated the contribution of traits loadings to each PCA axis using the ‘fviz_contrib’ function in the R package ‘factoextra’ (Fig S4).”

Fig. S4: Contribution of each trait loading to PCA axes. Plant height, seed mass and leaf dry matter content primarily contributed to PCA axis 1 (associated with plant size) among (a) global and (c) tundra species. LMA, leaf area and leaf dry matter content primarily contributed to PCA axis 2 (associated with resource) among (b) global and (d) tundra species. Bars indicate contribution of each trait to PCA axis; dashed red line indicates percentage contribution of all traits contributed equally (16.7%).

(5) Lack of clarity in writing.

This could be a minor comment, but given the frequency of instances when clarifications are needed, it significantly takes away from the ability to understand the manuscript. Here are the sentences that need clarification:

5.1- P6 ln7 “... and if relationships are confounded by trait variation within species”. I don’t know what that statement means. What relationships? Interspecific? Global? Do you mean if the interspecific relationships are affected by the intraspecific variation?

Amended to:

“Thus, it is unknown whether global trait relationships extend to climatic extremes, and if these interspecific relationships are confounded by trait variation within species.”

5.2- P6 Ins 10-11 “...Tundra plants demonstrated remarkable consistency in resource economic traits, but not size traits”. I am not sure what “consistency” means in this context. The same relationships as those found in other biomes? I am also unclear what is happening with size traits. What is not consistent

about them? If trait relationship in tundra plants are "not consistent" with those reported in global distributions, how can they exhibit the same two dimensions of trait variation?

Amended to:

"Tundra plants demonstrated remarkably similar resource economic traits, but not size traits, compared to global distributions."

5.3 - P6 Ins 12-13 "Three quarters of trait variation occurred among species, mirroring global estimates". Estimates of what? Of the extent of interspecific variation?

Amended to:

"Three quarters of trait variation occurred among species, mirroring global estimates of interspecific trait variation."

5.4 - P6 Ins 12-13 "Except at local scales" This is a confusing bit of information without further background on the question and methods. Does the importance of interspecific variation depend on the scale? If so, what do you mean by local? Save that result for somewhere where you have the space to explain what you mean. In the abstract, it is just confusing.

We have removed this line from the abstract.

5.5 - P6 Ins 14-15 "informing prediction of plant community change in a warming world". But the world is warming and all biomes will get warmer. So why is understanding relationships in the coldest biomes informative when all the biomes will get warmer? The world will include fewer arctic climates. The statement would be true if arctic biome were to become more common or colder, but it's the opposite.

We argue that understanding trait responses in this cold biome is critical for two main reasons:

1. The Arctic and alpine are warming more rapidly and more than the rest of the planet; this biome is a sentinel for global change. Understanding trait responses to warming in the tundra thus can inform our theoretical understanding of how plant communities elsewhere may respond to rapid temperature change. For example, whether trait trade-offs could constrain plant responses to change, whether trait axes respond equally to environmental pressures, or whether trait change is likely to occur within species, or require migration of new phenotypes to allow plant communities to track environmental change.
2. Understanding trait change in the tundra is itself a critical ecological question. The tundra biome represents the cold extreme of the temperature spectrum of life on planet earth. Even if these habitats are becoming warmer - and perhaps eventually decreasing in extent - the change that is happening here is a critical part of the overall climate response of our planet. In addition, vegetation feedbacks in the tundra, such as on decomposition and the carbon cycle, or on snow and albedo, have global implications. Such feedbacks are directly dependent upon trait change; understanding the rules underpinning trait expression and trait change in this previously underrepresented biome is thus of high priority.

5.6 - P6 In 34. "within species variation can invalidate trait relationships". This sentence is confusing. Specify which trait relationships you are talking about. Global interspecific? Also, what does "invalidate" mean? The relationships might be absent or reversed?

Amended to:

"Large within-species trait variation could obscure or alter interspecific trait relationships."

5.7 - P7 Ins 37-38: “within-species variation [...] has been hypothesized to be greater for species that span large biogeographical gradients or in extreme environments and at small geographical scales.” I can guess why that would be the case, but I shouldn’t have to be guessing. Please explain the rationale behind those arguments. The first and last scenario appear to be contradictory. Small spatial scales? For large biogeographical range, is it because of the large niche breadth (mechanism is adaptation to a wide variety of niches), or is it because of the species having many populations whose geographical isolation increase their ability to drift in different directions?

We agree that the original statements can be clarified and better set out underlying mechanisms of this relationship. There are a range of hypotheses concerning the relative importance of intraspecific variation across scales and in different types of environments, most of which are untested. Unfortunately we cannot fully cover the arguments here (though the references cited do provide several in depth discussions), and have therefore restructured this sentence to:

“Within-species variation accounts for approximately 25% of trait variation at the global scale (Siefert et al., 2015), but has been hypothesized to be greater in extreme environments due to environmental filtering of trait expression (Fajardo and Piper, 2011) at small geographical scales where species richness is low (Albert et al., 2011; Messier et al., 2017), and for species that span large biogeographical gradients (Siefert et al., 2015) due to wide niche breadth.”

5.8 - P7 In 39 replace “location to challenge” with “system to test”

Replaced.

5.9 - P7 In 53 “... yet will be constrained”. What will be constrained? The range of trait values ? Why this hypothesis? Provide a rationale for why would expect this hypothesis a priori. Is this like a null hypothesis?

Currently, it appears as if it as added a posteriori to the intro after finding this result. If we have no reason to hold this hypothesis, then just say that you are asking the question of how trait relationships differ in the tundra (although that has been answered in the two papers cited earlier) and state that we don't have expectations.

We have restructured the wording and order this sentence to make our hypothesis clearer and better reflect the previous discussion:

“1) Trait expression among tundra species will be constrained relative to global trait space due to extreme environmental conditions, yet will exhibit the same two dimensions of plant form and function.”

5.10 - P7 In56. Replace “relative to” with “than at”

Replaced.

5.11 - P7 In 59 “To examine constraints to trait expression”. Awkward wording. What does that mean? To examine trait ranges?

Amended to:

“To examine trait expression among tundra species”

5.12 - P7 Ins 66-67 “We tested whether sources of trait variation were dependent upon spatial scale”. I am not sure what you mean here. You mean you explored how your variance partitioning results change as your spatial scale increases?

Amended to:

“We tested how sources of trait variation varied across spatial scale”

5.13- P9 Ins 92-93 ““This broad range of resource economic trait values occurred independently of the bioclimatic range of tundra species”. I don't understand what that means. I read this as meaning that species with narrow or broad niche ranges all expressed a broad range of resource economic trait values? If that's what you mean, I have a hard time believing it.

This sentence was not clear and has now been amended to:

“Species with faster resource-related traits and larger size-related traits were also associated with warmer environments within the tundra (Fig S1).”

5.14- P9 In 100. Replace “partially constrained trait expression” with “showing a limited range of trait values”

Replaced.

5.15- P9 In 104. Add “relative to the global data” after “was reversed”

Added.

5.16- p10 In 123. Replace “principles” with “assumptions”

Replaced.

5.17- p10 In 125. Replace “estimates” with “means”

Amended to:

“was surprisingly close to the global mean”

5.18- p10 In 126. “large species ranges in the tundra (Fig 3b)”. Do tundra species have larger environmental niche or geographic ranges (spatial extent covered) than species in other biomes? I don't see how Figure 3b illustrates this. It shows the coefficient of variation for each trait (how variable trait values are relative to one another).

Removed reference to 3b.

5.19- P10 In 127. Replace “varied substantially by traits” with “changed substantially among traits”. “variation varied” is confusing.

Replaced.

5.20- P10 In 129. Is 25.6 the average contribution of functional group to total variance across all 6 traits? Why do you call 25% "little"? It seems to be as much as species-level variation for LMA and LDMC.

We have clarified the sentence to emphasise differences among traits:

“Functional group categorisation alone explained an average of 25.6% of trait variation across all six traits, and but less than 15% of variation for seed mass and plant height”

5.21- P10 In 130 delete “biogeographic”

Deleted.

5.22- P10 Ins 132-133. "species-level variation comprises the majority of the global spectrum...". This is only true for all your traits if you consider that the total species-level variation is the among-functional group + within-functional group species differences.

If you use a strict definition of species-level, (as I think you used in your analyses - i.e. within functional group – this circles back to my previous comment asking for this clarification), then this statement is only true for 3/6 of your traits.

I encourage the authors to change the legend and ms terminology to specify that "among species" refers to "among-species within functional groups".

We agree that Figure 3 alone is insufficient to support this statement due to the nested approach to analysis. However, the removal of functional group classifications (as in Figure 4) does not change this finding, and we argue that despite the nested nature of the approach our statement is still valid.

We have reordered the text, added reference to Figure 4, and added some additional explanatory text to address this point (L134-140).

"Differences among species explained the majority of trait variation in the tundra biome, accounting for an average of 76.8% of variation across the six traits examined (Fig 3a, Fig 4) and reinforcing one of the key assumptions of trait-based ecology (Shipley et al., 2016). Functional group categorisation alone explained an average of 25.6% of trait variation across all six traits, and but less than 15% of variation for seed mass and plant height (Thomas et al., 2019); differences among species still accounted for the majority of trait variation even if functional group classifications were removed. The contribution of within-species variation to total trait variation (23.2%) was surprisingly close to the global mean (25% (Siefert et al., 2015)), despite harsh environmental conditions and large species ranges in the tundra."

5.23- P10 In 133. Delete "global" and add "in tundra plants" after "function". Your results are for your TUNDRA plants. Stick to conclusions that your data can actually support...

The findings apply to tundra plants, but more broadly support the hypothesis that global trait variation (among these six traits) is primarily driven by species-level differences. We have altered the wording to make clear that our findings support this broader hypothesis, rather than indicating that what is true for the tundra is true everywhere.

"Overall, our findings support the hypothesis that species-level variation comprises the majority of the global spectrum of plant form and function."

5.24- P10-11 Ins 133-135. "underlining the importance of species richness and turnover in determining plant community characteristics, trait diversity, and linkages to ecosystem function.". This statement goes in the discussion section, not the results section.

The manuscript was written with an integrated results and discussion, with interpretation of individual results here, and overall synthesis and implications in the discussion section. Following the reviewers' suggestions, we have now moved the majority of such interpretation points to the discussion section (e.g. see similar point below). However, we argue that this particular interpretation is better placed here since it relates specifically to the role of ITV, rather than more broadly to the overall interpretations outlined in the discussion section. However, we will defer to the editors on whether our approach is appropriate in this instance.

5.25- P11 In151. Delete "clear scaling" (we don't know what you mean by scaling), add "is the first to" after 'study', and add 'between spatial scale and extent of intraspecific variation' after "relationship".

Amended.

5.26 – P12 In 156. This is true for 3 of the 4 traits, but not for N. You need to specify that to be correct.

Amended to 'in most traits'

5.27 - P13 Ins 167 "including resource acquisition, survival, competition and reproduction." This is a very broad claim. It would help if you reminded us of which trait is associated with which of these properties. E.g. "resource acquisition (LMA,LNC,LDMC), survival (?), competition (plant height) and reproduction (seed size).

A great idea. We have edited the text and included additional references to support links between traits and function.:

"Our findings reinforce claims that relationships between these widely measured plant traits are indicative of fundamental trade-offs in plant life strategy(McGill et al., 2006; Díaz et al., 2016), including resource acquisition (LMA, LA, LN, LDMC) (Reich, 2014), competition (PH, SM, LA, LDMC) (Kunstler et al., 2016), and reproduction (PH, SM) (Westoby, 1998)."

5.28 - P13 In 168 "conceptualization". I don't know what that means in the context of this sentence. The measurement of? I think the sentence would be clearer with no loss of information if you just delete this bit and write "..., trait space should include seldomly measured traits..."

Amended.

5.29 - P13 In 169. "underrepresented traits". do you just mean traits you haven't measured in this study? Or do you mean traits that are never measured? It would be more informative if you were more specific than simply stating that we need to measure more traits. Then the comment "that capture critical links to ecosystem function". You are opening a bag of worms here. Are you saying your traits have poor link to ecosystem function? That was part of the justification for measuring them earlier... Also, do we know that below-ground traits have stronger links to ecosystem function? Which traits? which function? This sentence is so vague that it does not bring any value. Either delete it or state something more specific.

Amended to:

"However, there remains a need to integrate plant size and resource economics with other key facets of plant life strategy such as phenology (Bjorkman et al., 2017), and to understand how well they organise rarely-measured traits such as chemical and below-ground traits (Iversen et al., 2015; Myers-Smith et al., 2019b)."

Other Minor comments

P7 In 47 "89% of the tundra species pool". That's great coverage. What kind of replication do you get for each species? Multiple populations? Multiple individuals? TRY is a bit infamous for reporting trait means.

The dataset used in this study was built over a large number of years, with around 60% of individual trait values directly submitted by co-authors, and the remaining 40% compiled from TRY data. Where species' means were used, they were checked to make sure there was no overlap with individual trait measurements (in this case, mean values were excluded).

In the data used here, each species had a mean of 114 observations and 8 unique locations. 32 species had only one trait observation, while the most common species had the most observations and sites, as might be expected. Species with a low number of observations and sites (i.e. fewer than three) were excluded from variance partitioning analysis. See also Table S1.

A detailed description of the full TTT dataset can be found in Bjorkman et al. (2018).

P7 Ins 63-66 Are species nested within each functional group, or does the species scale go across functional groups? in the later case?

Species are nested within functional groups.

P6 In 67 replace “were dependent upon” with ‘depend on the’. Using active verbs is always clearer.

Replaced.

P8 Figure 1 panel C. replace “Observations” in y-axis label of upper panel with “Trait observations” Replaced.

P9 In 94. Figure S1 seems to contradict Figure 2B, where the trait hypervolume occupied by cold and warm species is smaller than the hypervolume occupied by mid-temp species. Imagine drawing convex hulls around each of these 3 species group. The mid-temp species clearly would cover a much larger area than the warm and cold species. Where am I going wrong and how can we avoid such confusion on the part of the reader?

We have re-run the within-tundra analyses (Figure S1) in light of these comments and additional suggestions from reviewer 1. These now more clearly show that

1. At the species level, there is slight increase in size traits from ‘cold’ to ‘warm’ categories (noting that this is at the species level, and does not reflect within-species variation across biogeographical gradients).
2. At the species level, there is a clear increase in resource traits from ‘cold’ to ‘warm’ categories among species.

P9 Ins 95-97. “tundra, this variation in plant leaf resource economics is remarkably high, and suggests that plants have developed a wide range of ecological strategies to cope with extreme conditions”. To me, this is the that’s the most interesting finding. I would write the paper around that result instead of around asking whether trait relationships extend to the arctic, which we know they did from Reich’s work.

We have now restructured and slightly expanded the discussion section to place greater emphasis on this finding and add greater clarity.

P9 Ins108-110. Move this statement to the discussion section. This is an interpretation, not a result. Amended.

P13 In 175 In line with my previous comments, replace “that the two major axes” with “that two of the major axes”

Replaced.

P13 In 186. Replace “inform” with “support”.

Replaced.

P14 In195. “better incorporate hierarchical trait variation”. This statement makes me wonder how hierarchical trait variation is currently specified in those models, and how you propose to improve it.

Trait variation – within communities and across environmental gradients - is important for 1) understanding plasticity, 2) capturing uncertainty, and 3) understanding trait-climate relationships. However, as the reviewer rightly recognises, trait-based approaches often do

not take trait variation into account, for example using trait, species or site means, or collecting a very small number of measurements. Similarly, detailed information on collection sites is not always recorded, making it difficult to link trait values to the environmental conditions in which they were collected, and thus impossible to examine variation and change across environmental gradients.

We hope that this study adds further weight to those proposing greater incorporation of trait variation into analyses by demonstrating that 1) although the greatest trait differences occur among species, within-species variation is still significant, 2) the extent of trait variation is highly trait dependent, and 3) sources of trait variation are also scale-dependent.

We therefore particularly welcome studies that explicitly seek to incorporate trait variation into analyses. These include Suding et al., (2008), Soudzilovskaia et al., (2013), Bjorkman et al., (2018), among others. We have clarified that we view these studies as examples of best practice, and that for such approaches to be more widely used trait data (and databases) need to better link individual records to potential sources of variation.

We have thus revised the text as follows (L212-217):

“Quantifying variation across spatial and temporal scales has been shown to constrain trait-based vegetation models (Lavorel and Garnier, 2002; Reich, 2014) and subsequently improve prediction of the response of key ecosystem processes to environmental change (Suding et al., 2008; Soudzilovskaia et al., 2013; Bjorkman et al., 2018). However, such trait-based modelling approaches are rare, and require precisely geo-referenced trait databases that link trait records to environmental variables.”

Reviewers' comments:

Reviewer #1 (Remarks to the Author):

I appreciated the thoroughness with which the authors addressed by comments on the previous version of the manuscript. The additional panels in Fig 2 were helpful and I would recommend including them. It was also good to have the species in the extreme of the trait space identified. The discussion on trees is obviously a much bigger topic than is covered by the scope of this paper but the additional text was helpful in acknowledging its relevance to interpreting the results. However, I was not convince that the additional figure distinguishing trees in the traits space was necessary.

Reviewer #3 (Remarks to the Author):

GENERAL COMMENTS

- This study evaluates whether global trait relationships can be extended to the coldest extremes of life on Earth (tundra biomes) using six broadly-explored aboveground traits. They found that tundra plant traits were aligned along the same two dimensions of trait variation detected at the global level (i.e. plant size and leaf economics spectrum). They also partitioned the sources of variation of the six studied plant traits, and found that the contribution of within-species variation was particularly low in the tundra ecosystems in comparison with the global average of 25%.

- Strengths:

1. The authors used an impressive data set.
2. The manuscript is clear and well-written.
3. The topic is potentially interesting for a large number of readers of Nature Communications.

- Weaknesses:

1. I have found some problems with the structure of the manuscript. For example, the "Results" section includes some inferences that are more appropriate for the "Discussion" section (e.g. lines 93-99, 120-123, 165-168, etc.). I recommend the authors to move this information to the "Discussion" or to join these two parts in a "Results and Discussion" section.

2. The Discussion is too concise and poor in references. There are some results that have been barely discussed.

First, I have missed some potential hypotheses in this section for explaining why the contribution of within-species trait variation in the tundra was particularly low in comparison with the global average. I think that there are two possibilities: (i) there is a high functional divergence among species (and, thereby, a large contribution of this source of trait variability); or (ii) tundra species exhibit a low phenotypic plasticity in these morphological traits. The first explanation is partially supported by the large variation in plant resource economics that was detected in this biome, suggesting that tundra species have developed a wide range of ecological strategies to cope with extreme conditions and limiting resources. However, this result does not preclude the second option, although the authors should recognize specifically in the text that tundra species were not highly plastic in their morphology but this does not imply that they can exhibit a large intraspecific variability in other rarely-measured traits that were not considered in this study. Previous studies have shown that other traits that are not related with plant aboveground morphology (e.g. root, physiological or phenological traits), could be involved in some ecological processes that drive species coexistence under particular environmental conditions. For example, in a recent study published in *Nature Communications*, Pérez-Ramos et al. (2019) demonstrated that the competitive ability of plant species in Mediterranean grasslands is sometimes more dependent on their plasticity in certain physiological traits related to water-use ($\delta^{13}\text{C}$), photosynthetic (A_{max} , g_s) and light-use (light curve convexity) efficiency than on morphological traits commonly associated to ecological resource-use strategies.

Second, the result showing that within-species trait variation contributed to a greater proportion of total trait variation at small geographical scales has not been discussed enough. The authors should also consider other previous studies reporting that some patterns of trait covariation can become weaker or even disappear when considering groups of species belonging to environmentally similar sites (e.g. de la Riva et al. 2016).

Finally, the contribution of functional groups to the total trait variation was relatively high for some traits, such as LDMC or leaf area, but this result has not been discussed in the new version of the manuscript. For instance, it would be interesting to know how many types of functional groups are identified in the tundra in order to know if the range of ecological strategies that are present in this biome is restricted in comparison with those present at higher geographical scales.

MINOR COMMENTS

- Line 60: Hypotheses 2 and 3 are not referred to the magnitude of within-species trait variation but to its relative contribution to total trait variation, does not it? Correct this, please.
- Line 138: Delete “and” just before the word “but”.
- Line 145: greater variation within species? Specify, please.
- Line 166: I imagine the authors want to say that “the contribution of within-species trait variation..”
- Line 181: Replace “subject” by “subjected”.

- Line 186: It is not clear what plant traits are implied in competition. See my comment above.
- Line 208: To improve technology?? It is not clear what the authors want to say with this sentence. Rephrase it, please.

REFERENCES CITED

de la Riva, E. G., Tosto, A., Pérez-Ramos, I.M., Navarro-Fernández, C.M., Olmo, M., Anten, N.P.R., Marañón, T. & Villar, R. (2016) A plant economics spectrum in Mediterranean forests along environmental gradients: is there coordination among leaf, stem and root traits? *Journal of Vegetation Science* 27: 187–199

Pérez-Ramos, I. M., Matías, L., Gómez-Aparicio, L. & Godoy, O. (2019). Functional traits and phenotypic plasticity modulate species coexistence in changing environments. *Nature Communications*

SCHOOL of GEOSCIENCES The
University of Edinburgh Crew
Building, Kings Buildings
Edinburgh EH9 3FF United
Kingdom

hidthomas@gmail.com

3rd January 2020

Dear reviewers,

Thank you for your further comments and feedback on our manuscript, 'Global plant trait relationships extend to the climatic extremes of the tundra biome'. We have now completed our revisions and hope that we have been able to address all of your comments and concerns. We have made the following major changes, in addition to smaller improvements and formatting changes throughout the text:

1. Reorganisation of the discussion and addition of two new paragraphs to address Reviewer 3's concern that the contribution and implications of within-special trait variation is not adequately recognised.
2. Further contextualisation and reference to the literature, including non-tundra context as suggested by Reviewer 3.
3. Inclusion of additional data panels showing 'extreme' tundra analyses within Figure 2, following Reviewer 1's request.
4. Reformatting of the text in line with journal requirements, including moving key methods from supplementary information to main text.

All changes are outlined in detailed point-by-point responses below, and are highlighted in blue within the main manuscript file.

Yours sincerely,

Haydn Thomas,
On behalf of all authors.

Reviewers' comments:

Reviewer #1 (Remarks to the Author):

I appreciated the thoroughness with which the authors addressed by comments on the previous version of the manuscript. The additional panels in Fig 2 were helpful and I would recommend including them. It was also good to have the species in the extreme of the trait space identified. The discussion on trees is obviously a much bigger topic than is covered by the scope of this paper but the additional text was helpful in acknowledging its relevance to interpreting the results. However, I was not convince that the additional figure distinguishing trees in the traits space was necessary.

We thank the reviewer for providing this follow-up review. We have included the additional two panels within Figure 2 in the main text. We retain the additional text on trees but have not included the figure, as suggested. We have expanded the figure caption, and methods in the main text and supplementary information to reflect these additional panels.

Reviewer #3 (Remarks to the Author):

GENERAL COMMENTS

- This study evaluates whether global trait relationships can be extended to the coldest extremes of life on Earth (tundra biomes) using six broadly-explored aboveground traits. They found that tundra plant traits were aligned along the same two dimensions of trait variation detected at the global level (i.e. plant size and leaf economics spectrum). They also partitioned the sources of variation of the six studied plant traits, and found that the contribution of within-species variation was particularly low in the tundra ecosystems in comparison with the global average of 25%.

- Strengths:

1. The authors used an impressive data set.
2. The manuscript is clear and well-written.
3. The topic is potentially interesting for a large number of readers of Nature Communications.

- Weaknesses:

1. I have found some problems with the structure of the manuscript. For example, the “Results” section includes some inferences that are more appropriate for the “Discussion” section (e.g. lines 93-

99, 120-123, 165-168, etc.). I recommend the authors to move this information to the “Discussion” or to join these two parts in a “Results and Discussion” section.

We agree with this point. We would prefer to keep the current structure as we believe that it provides a clearer summary of the main findings. As such, we now present this section as a “Results and Discussion” section following the reviewer’s suggestion.

2. The Discussion is too concise and poor in references. There are some results that have been barely discussed.

We thank the reviewer for these comments and very much welcome the opportunity to expand upon these topics in the discussion, particularly with respect to the contribution of within-species trait variation. We have now restructured and expanded the discussion to address these points, and included additional references. We address each of the points raised by the reviewer below:

First, I have missed some potential hypotheses in this section for explaining why the contribution of within-species trait variation in the tundra was particularly low in comparison with the global average. I think that there are two possibilities: (i) there is a high functional divergence among species (and, thereby, a large contribution of this source of trait variability); or (ii) tundra species exhibit a low phenotypic plasticity in these morphological traits. The first explanation is partially supported by the large variation in plant resource economics that was detected in this biome, suggesting that tundra species have developed a wide range of ecological strategies to cope with extreme conditions and limiting resources. However, this result does not preclude the second option, although the authors should recognize specifically in the text that tundra species were not highly plastic in their morphology but this does not imply that they can exhibit a large intraspecific variability in other rarely-measured traits that were not considered in this study.

We now discuss how there could be two different explanations for why the contribution of within-species trait variation in the tundra was particularly low in comparison with the global average:

“Contrary to our expectations, the contribution of within-species trait variation in the tundra was lower than global estimates¹. Our findings support previous studies indicating that the relative importance of within-species variation decreases with increasing environmental stress², leading to wide functional divergence between species, as found for resource-economic traits^{3,4}. Lower within-species variation may further indicate that plasticity is lower among tundra species, which are typically slow growing and nutrient limited⁵. However, rapid and sustained plastic responses to environmental change have been documented at some tundra sites^{6,7}. Indeed, large differences in both trait expression, as demonstrated here, and plasticity have been found to promote coexistence in other resource-limited communities⁸. If the majority of trait variation occurs among species, and if phenotypic plasticity is comparatively low, shifts in community-level traits following environmental change may thus occur more slowly than would be predicted from biogeographic gradients^{9,10}. More substantial trait change would thus require the immigration of new species from warmer sites.” ”

Previous studies have shown that other traits that are not related with plant aboveground morphology (e.g. root, physiological or phenological traits), could be involved in some ecological processes that drive species coexistence under particular environmental conditions. For example, in a recent study published in *Nature Communications*, Pérez-Ramos et al. (2019) demonstrated that the competitive ability of plant species in Mediterranean grasslands is sometimes more dependent on their plasticity in certain physiological traits related to water-use ($\delta^{13}C$), photosynthetic (A_{max} , g_s) and light-use (light curve convexity) efficiency than on morphological traits commonly associated to ecological resource-use strategies.

We refer to the potential for other key traits to contribute to changing the global trait space occupied by tundra:

“However, plant size and resource economics have yet to be integrated with other key facets of plant life strategy such as phenology^{8,11}, chemical and below-ground traits^{12,13}. These less frequently measured traits need to be incorporated into analyses to more comprehensively capture how extreme biomes such as the tundra occupy global trait space.”

Second, the result showing that within-species trait variation contributed to a greater proportion of total trait variation at small geographical scales has not been discussed enough. The authors should also consider other previous studies reporting that some patterns of trait covariation can become weaker or even disappear when considering groups of species belonging to environmentally similar sites (e.g. de la Riva et al. 2016).

We now discuss within-species variation in the tundra in greater detail, with particular reference to changes over geographical scale, and contributions to trait variation at small geographical scales. We also note the importance of studies such as de la Riva et al. 2016 in demonstrating the importance of accounting for within-species variation, both in the discussion paragraph below and in a slightly revised introductory paragraph (relevant lines also copied below).

Discussion paragraph

“We found that within-species trait variation comprised a large component of trait variation at local scales – the scale at which many critical ecological processes occur¹⁴. Despite the importance of trait differences among species in the tundra, we nevertheless found that that within-species variation accounted for approximately one quarter of total trait variation, and thus should not be ignored in trait-based analyses¹⁵. High within-species trait variation at local scales has previously been predicted from ecological theory^{1,16}, and may result from low local-scale species richness¹⁶ or reveal the influence of local-scale environmental variability (i.e. topography, snow, drainage, etc)^{17–20}. Accounting for multiple sources of trait variation has been shown to constrain trait-based vegetation models^{21,22} and subsequently to improve prediction of the response of key ecosystem processes to environmental change^{9,14,23}. However, such trait-based modelling approaches are rare, and require precisely geo-referenced trait databases that link trait records to environmental variables. We therefore support calls to collect additional trait data in changing and novel climate conditions^{9,24}, to improve the techniques and technologies used to remotely sense plant trait information²⁵, and to incorporate trait variation into Earth system modelling²⁶.”

Introductory paragraph

“Our current understanding of global trait relationships is also based on the assumption that the majority of trait variation occurs among species²⁷. However, trait variation within communities is ultimately driven by differences among individuals, rather than species¹. Large within-species trait variation could thus obscure or alter interspecific trait relationships^{17,28,29}, restricting their potential for ecological prediction across scales and among biomes.”

Finally, the contribution of functional groups to the total trait variation was relatively high for some traits, such as LDMC or leaf area, but this result has not been discussed in the new version of the manuscript. For instance, it would be interesting to know how many types of functional groups are identified in the tundra in order to know if the range of ecological strategies that are present in this biome is restricted in comparison with those present at higher geographical scales.

Regarding this final comment, we have published a separate study (Thomas *et al.*, 2019, Global Ecology and Biogeography), which explores the role of functional groups within the tundra. The purpose of that study was to test whether, and how, functional groups capture trait differences among species in the tundra, and thus in which situations they may be best applied in ecological analyses. We found that (1) although the overall trait variation explained by tundra functional groups is relatively low, resource-economic traits (including LDMC) are well explained by groups; and (2) alternative, *post-hoc* groupings of species based on their traits (including using more than the four traditional functional groups) substantially increases the overall variation explained, including of both size and economic traits.

We refer to the Thomas *et al.* 2019 in the following sentences in the manuscript:

“Functional group categorisation alone explained an average of 25.6% of trait variation across all six traits, but varied substantially by trait³; differences among species still accounted for the majority of trait variation even if functional group classifications were removed.”

and

“Our findings support previous suggestions that the relative importance of within-species variation decreases with increasing environmental stress², leading to wide functional divergence between species, as found for resource-economic traits^{3,4}”

Although the focus of the functional groups study differs substantially from the questions being tested here, we discuss in some depth the issues raised by the reviewer in that manuscript. We therefore believe that the references to the previously published study in this manuscript should suffice.

MINOR COMMENTS

- Line 60: Hypotheses 2 and 3 are not referred to the magnitude of within-species trait variation but to its relative contribution to total trait variation, does not it? Correct this, please.

We have corrected these to:

“2) The contribution of within-species trait variation to total trait variation in the tundra will be greater than the global average of 25%. 3) The contribution of within-species trait variation to total trait variation will be greater at small geographical scales than at large scales.”

- Line 138: Delete “and” just before the word “but”.

Deleted.

- Line 145: greater variation within species? Specify, please.

We have corrected this to:

“Size-related traits demonstrated greater overall variation than resource economic traits, even though variation relative to global trait space was constrained along the size-related axis”

- Line 166: I imagine the authors want to say that “the contribution of within-species trait variation..”

Corrected as suggested.

- Line 181: Replace “subject” by “subjected”.

Replaced.

- Line 186: It is not clear what plant traits are implied in competition. See my comment above.

We believe this comment is now covered by the revised discussion, in addition to the traits and references included here for clarity.

- Line 208: To improve technology?? It is not clear what the authors want to say with this sentence. Rephrase it, please.

We have rephrased this to read:

“More broadly, our findings support calls to collect additional trait data in changing and novel climate conditions, to improve the techniques and technologies used to remotely sense plant trait information, and to incorporate trait variation into Earth system modelling.”

[REVIEWER] REFERENCES CITED

- de la Riva, E. G., Tosto, A., Pérez-Ramos, I.M., Navarro-Fernández, C.M., Olmo, M., Anten, N.P.R., Marañón, T. & Villar, R. (2016) A plant economics spectrum in Mediterranean forests along environmental gradients: is there coordination among leaf, stem and root traits? *Journal of Vegetation Science* **27**: 187–199
- Pérez-Ramos, I. M., Matías, L., Gómez-Aparicio, L. & Godoy, O. (2019). Functional traits and phenotypic plasticity modulate species coexistence in changing environments. *Nature Communications*

[Response] References cited

1. Siefert, A. *et al.* A global meta-analysis of the relative extent of intraspecific trait variation in plant communities. *Ecology Letters* **18**, 1406–1419 (2015).
2. Fajardo, A. & Piper, F. I. Intraspecific trait variation and covariation in a widespread tree species (*Nothofagus pumilio*) in southern Chile. *New Phytologist* **189**, 259–271 (2011).
3. Thomas, H. J. D. *et al.* Traditional plant functional groups explain variation in economic but not size-related traits across the tundra biome. *Global Ecology and Biogeography* **28**, 78–95 (2019).
4. Hoffmann, A. A. & Merilä, J. Heritable variation and evolution under favourable and unfavourable conditions. *Trends in Ecology and Evolution* **14**, 96–101 (1999).
5. Baruah, G., Molau, U., Bai, Y. & Alatalo, J. M. Community and species-specific responses of plant traits to 23 years of experimental warming across subarctic tundra plant communities. *Scientific Reports* **7**, 2571 (2017).
6. Hudson, J. M. G., Henry, G. H. R. & Cornwell, W. K. Taller and larger: shifts in Arctic tundra leaf traits after 16 years of experimental warming. *Global Change Biology* **17**, 1013–1021 (2011).
7. Myers-Smith, I. H. *et al.* Eighteen years of ecological monitoring reveals multiple lines of evidence for tundra vegetation change. *Ecological Monographs* **89**, (2019).
8. Pérez-Ramos, I. M., Matías, L., Gómez-Aparicio, L. & Godoy, Ó. Functional traits and phenotypic plasticity modulate species coexistence across contrasting climatic conditions. *Nature Communications* **10**, (2019).
9. Bjorkman, A. D. *et al.* Plant functional trait change across a warming tundra biome. *Nature* **562**, 57–62 (2018).
10. Elmendorf, S. C. *et al.* Experiment, monitoring, and gradient methods used to infer climate change effects on plant communities yield consistent patterns. *Proceedings of the National Academy of Sciences* **112**, 448–452 (2015).
11. Bjorkman, A. D., Vellend, M., Frei, E. R. & Henry, G. H. R. Climate adaptation is not enough: warming does not facilitate success of southern tundra plant populations in the high Arctic. *Global Change Biology* **23**, 1540–1551 (2017).
12. Iversen, C. M. *et al.* The unseen iceberg: plant roots in arctic tundra. *New Phytologist* **205**, 34–58 (2015).
13. Myers-Smith, I. H., Thomas, H. J. D. & Bjorkman, A. D. Plant traits inform predictions of tundra responses to global change. *New Phytologist* **221**, 1742–1748 (2019).
14. Suding, K. N. *et al.* Scaling environmental change through the community-level: a trait-based response-and-effect framework for plants. *Global Change Biology* **14**, 1125–1140 (2008).
15. Violle, C. *et al.* The return of the variance: Intraspecific variability in community ecology. *Trends in Ecology and Evolution* **27**, 244–252 (2012).
16. Albert, C. H., Grassein, F., Schurr, F. M., Vieilledent, G. & Violle, C. When and how should intraspecific variability be considered in trait-based plant ecology? *Perspectives in Plant Ecology, Evolution and Systematics* **13**, 217–225 (2011).
17. De La Riva, E. G., Olmo, M., Poorter, H., Ubersa, J. L. & Villar, R. Leaf mass per area (LMA) and its relationship with leaf structure and anatomy in 34 mediterranean woody species along a water availability gradient. *PLoS ONE* **11**, (2016).
18. Opedal, Ø. H., Armbruster, W. S. & Graae, B. J. Linking small-scale topography with microclimate, plant species diversity and intra-specific trait variation in an alpine landscape. *Plant Ecology and Diversity* **8**, 305–315 (2015).
19. Elberling, B. Annual soil CO₂ effluxes in the High Arctic: The role of snow thickness and vegetation type. *Soil Biology and Biochemistry* **39**, 646–654 (2007).
20. McGraw, J. B. Experimental ecology of *Dryas octopetala* ecotypes. III. Environmental factors and plant growth. *Arctic & Alpine Research* **17**, 229–239 (1985).

21. Reich, P. B. The world-wide ‘fast-slow’ plant economics spectrum: A traits manifesto. *Journal of Ecology* **102**, 275–301 (2014).
22. Lavorel, S. & Garnier, E. Predicting changes in community composition and ecosystem functioning from plant traits: revisiting the Holy Grail. *Functional Ecology* **16**, 545–556 (2002).
23. Soudzilovskaia, N. A. *et al.* Functional traits predict relationship between plant abundance dynamic and long-term climate warming. *Proceedings of the National Academy of Sciences* **110**, 18180–18184 (2013).
24. Bjorkman, A. D. *et al.* Tundra Trait Team: A database of plant traits spanning the tundra biome. *Global Ecology and Biogeography* **27**, 1402–1411 (2018).
25. Jetz, W. *et al.* Monitoring plant functional diversity from space. *Nature Plants* **2**, 1–5 (2016).
26. Wullschlegel, S. D. *et al.* Plant functional types in Earth system models: Past experiences and future directions for application of dynamic vegetation models in high-latitude ecosystems. *Annals of Botany* **114**, 1–16 (2014).
27. Shipley, B. *et al.* Reinforcing loose foundation stones in trait-based plant ecology. *Oecologia* 1–9 (2016). doi:10.1007/s00442-016-3549-x
28. Anderegg, L. D. L. *et al.* Within-species patterns challenge our understanding of the leaf economics spectrum. *Ecology Letters* **21**, 734–744 (2018).
29. Laughlin, D. C. *et al.* Intraspecific trait variation can weaken interspecific trait correlations when assessing the whole-plant economic spectrum. *Ecology and Evolution* **7**, 8936–8949 (2017).